# Resolving the Timestep Scaling Paradox in Spiking Neural Networks with a Timestep-Scalable Neuron Model

**Binghao Ye** [* 1 2] **Wenjuan Li** [* 3 4] **Dengfeng Xue** [5] **Bing Li** [3 4 6 7] **Weiming Hu** [3 4 6 8] **Dong Liang** [1 9] **Kun Shang** [1 9]

## Abstract

Spiking Neural Networks (SNNs) have garnered attention for their biological plausibility, energy efficiency, and temporal modeling capability. Due to the non-differentiability of spike generation, a widely used training method for SNNs is back-propagation through time with surrogate gradients, achieving competitive performance with few timesteps. Intuitively, scaling timesteps should improve performance by enriching temporal dynamics. However, we observe timestep scaling paradox (TSP), a counter-intuitive accuracy degradation when scaling timesteps. We investigate TSP and link it to long-term temporal gradient vanishing and weakened cross-timestep dependencies. To address this, we propose the Timestep-Scalable (TS) neuron model. It introduces long-term memory reconsolidation to enhance cross-timestep information flow and enable effective learning with more timesteps. In parallel, a temporal forgetting mechanism periodically truncates the accumulation path, suppressing excessive temporal buildup and improving training stability. Supported by theoretical analysis and extensive experiments, TS consistently improves performance when scaling timesteps. In addition, it attains state-of-the-art results on time-series and event-based tasks, while remaining strong on static image classification and object detection.

---

[*]Equal contribution [1]Shenzhen Institutes of Advanced Technology, Chinese Academy of Sciences [2]University of Chinese Academy of Sciences [3]State Key Laboratory of Multimodal Artificial Intelligence Systems, CASIA [4]Beijing Key Laboratory of Super Intelligent Security of Multi-Modal Information [5]School of Computer Science and Technology, Xidian University [6]School of Artificial Intelligence, University of Chinese Academy of Sciences [7]PeopleAI Inc. [8]School of Information Science and Technology, ShanghaiTech University [9]Guangdong Provincial Key Laboratory of Multimodality Non-Invasive Brain-Computer Interfaces. Correspondence to: Dong Liang <dong.liang@siat.ac.cn>, Kun Shang <kunzzz.shang@gmail.com>.

*Proceedings of the 43$^{rd}$ International Conference on Machine Learning*, Seoul, South Korea. PMLR 306, 2026. Copyright 2026 by the author(s).

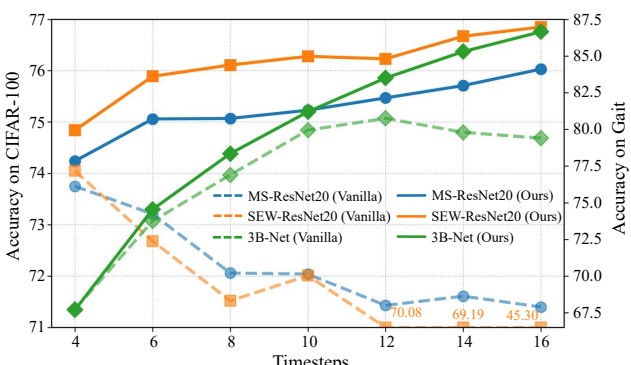

*Figure 1.* Performance trends as the number of timesteps increases for SEW-ResNet20 and MS-ResNet20 on CIFAR-100, and for 3B-Net on the Gait dataset. Our method scales favorably with longer timesteps, yielding stable and consistently improving performance as timesteps increase, whereas vanilla SNNs often suffer performance degradation at larger timesteps.

## 1. Introduction

Spiking Neural Networks (SNNs) have attracted growing attention for their brain-inspired information processing mechanism (Zhao et al., 2025; Yu et al., 2025b; Ding et al., 2025; Hwang et al., 2024; Yao et al., 2025) and are regarded as the third generation of neural networks (Maass, 1997). In SNNs, information is conveyed through binary spike trains, where neuronal membrane potentials integrate incoming signals over time and emit a spike upon reaching a threshold, otherwise remaining silent. This enables event-driven computation and allows SNNs to replace a large number of multiply-accumulate operations (MACs) with cheaper sparse accumulate operations (ACs), thereby offering significantly improved energy efficiency compared with artificial neural networks (ANNs) in neuromorphic chips (Davies et al., 2018; Ma et al., 2017; Yao et al., 2024b).

However, supervised training of SNNs is challenging due to the non-differentiability of spike generation. A mainstream approach is backpropagation through time (BPTT) with surrogate gradients (SG) (Werbos, 2002; Cramer et al., 2022), which replaces the derivative of the spike-generation function with a surrogate gradient during backpropagation. This enables gradients to propagate along both temporal

and spatial dimensions, and achieves competitive performance with only a small number of timesteps (a short spike train) (Guo et al., 2024b; Yu et al., 2025a). In principle, increasing the number of timesteps should improve performance by enabling richer temporal dynamics. However, as shown in Figure 1, we observe the opposite in practice, which we refer to as timestep scaling paradox (TSP). On temporal datasets, performance improves only marginally or even degrades as the number of timesteps increases (results for larger timesteps are provided in Section 5.1), and the degradation is typically more pronounced on static datasets. Similar observations have been reported in multiple studies (Meng et al., 2023; Yao et al., 2021; Hu et al., 2024a), yet the underlying cause is rarely examined in depth. To fill this gap, we analyze TSP and find that it is tied to the limited intrinsic complexity of the conventional spiking neuron. This limitation causes temporal gradients to vanish over long timesteps, making it difficult for error signals from later timesteps to propagate back to earlier neuronal states and weakening cross-timestep dependencies. In response, some prior works view temporal gradients during backpropagation as less influential, and thus ignore them in the backward pass to simplify optimization (Meng et al., 2023; Xiao et al., 2022). However, we view temporal gradient propagation and cross-timestep dependencies as key advantages of SNNs, so they should not be discarded.

In this paper, we aim to improve temporal gradient propagation and cross-timestep dependencies by increasing the intrinsic complexity of the spiking neuron. To this end, we propose the Timestep-Scalable Leaky Integrate and Fire (TS-LIF) neuron model, which consists of long-term memory reconsolidation and a temporal forgetting mechanism. Long-term memory reconsolidation injects earlier membrane potentials into the neuronal state update, providing an additional pathway for information and gradients to propagate across timesteps. The temporal forgetting mechanism periodically truncates the accumulation path to prevent excessive temporal buildup and stabilize long-term training. We further apply normalization to prevent scale drift and stabilize membrane dynamics. Finally, we provide theoretical and experimental analyses to characterize temporal gradient propagation under our design, demonstrating that it effectively mitigates long-term temporal gradient vanishing. Our method enables SNNs to consistently benefit from increasing the number of timesteps, leading to sustained performance gains, and achieves state-of-the-art results across multiple datasets. Overall, our main contributions are:

- We present the first in-depth analysis of TSP and trace it primarily to long-term temporal gradient vanishing. Thus, we propose TS-LIF model, which combines long-term memory reconsolidation and a temporal forgetting mechanism to strengthen and stabilize temporal

gradient propagation. With our design, SNNs can reliably benefit from increasing timesteps, providing a new pathway for improving SNN performance.

- We provide theoretical analysis and experiments to validate that TS-LIF mitigates temporal gradient vanishing and increases the contribution of temporal gradients, thereby strengthening cross-timestep dependencies.

- Our method achieves strong performance across time-series and event-based datasets, as well as conventional static datasets for image classification and object detection, and generalizes well across different architectures such as ResNet and SpikeYOLO. For example, on Sleep-EDF, our model achieves 77.20% accuracy, outperforming the baseline by 6.3%.

## 2. Related Work

**SNNs Training with BPTT.** A mainstream approach to training SNNs is BPTT with SG. Since the spike generation function is non-differentiable, SG replaces its intractable derivative with a surrogate derivative during the backward pass (Neftci et al., 2019). Under this framework, gradients are propagated not only through network layers but also along the temporal unfolding of neuronal states (Wu et al., 2018). This training method has demonstrated strong empirical performance across a wide range of tasks under short timestep settings (Lei et al., 2025; Miao et al., 2025; Li et al., 2025; Sun et al., 2024; Xue et al., 2026; Wan et al., 2026). Moreover, backpropagation through time with surrogate gradients enables the direct training of deeper SNN architectures, which achieve higher performance (Hu et al., 2024b; Fang et al., 2021). In contrast, the timestep dimension has received far less systematic investigation, leaving the potential of timestep scaling underexplored. In principle, increasing the number of timesteps should enable longer-term integration of temporal information and improve accuracy. Surprisingly, we observe TSP, a performance degradation when scaling timesteps. In this work, we analyze and address this issue, providing a practical pathway for improving SNN performance via temporal scaling.

**Alternative Training Method.** ANN-to-SNN conversion is another route to obtain high-performing SNNs (Hao et al., 2023a; Li et al., 2021a; Rueckauer et al., 2017; Huang et al., 2025; 2024). It starts from a pre-trained ANN and replaces its activation functions with spiking neurons, while calibrating the spiking behavior such that the firing rates match the ANN activations. However, achieving performance comparable to the original ANN typically requires a large number of timesteps (Deng & Gu, 2021; Han & Roy, 2020; Wang et al., 2023). Moreover, this approach ignores the temporal dynamics of SNNs and can limit its effectiveness on neuromorphic datasets. Another approach is the biologically

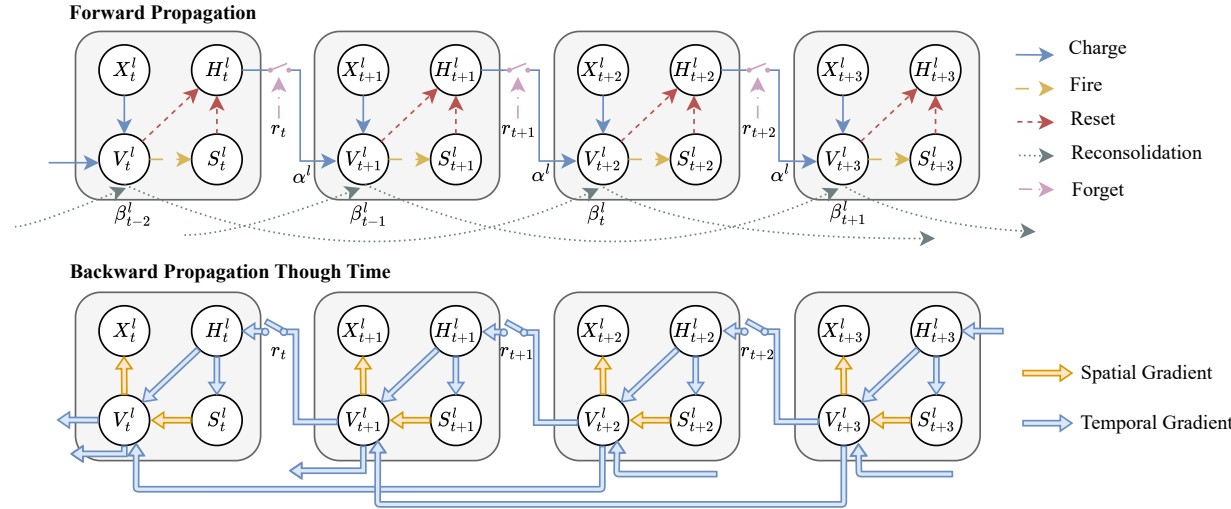

*Figure 2.* Overview of the forward and backward propagation of TS-LIF. For clarity, we illustrate $t > 2$ in the figure to better depict the workflow. Subscripts in the figure denote timesteps and are equivalent to the bracket notation $[t]$ used in the text.

inspired spike-timing-dependent plasticity (STDP) (Bi & Poo, 1998). Following Hebbian learning principles (Hebb, 2005), STDP updates synaptic weights in an unsupervised manner. However, it typically yields inferior performance and is usually limited to small-scale datasets.

**Temporal Gradient Issues in SNNs.** Temporal-gradient issues are widely observed in conventional SNN training. SLTT (Meng et al., 2023) reports that gradients propagated along the temporal dimension can be negligible under BPTT-style training and thus proposes ignoring temporal dependencies during optimization to simplify training while retaining comparable accuracy. OTTT (Xiao et al., 2022) enables online learning by decoupling the temporal dependency in BPTT, which likewise removes certain temporal dependency paths in the backward pass. Such designs can undermine the ability of SNNs to model temporal dynamics, a key characteristic of SNNs. DeepTAGE (Liu et al., 2025) observes that gradient magnitudes tend to diminish as time progresses. It adaptively adjusts the surrogate gradient at each timestep based on the membrane potential distribution, and incorporates spatio-temporal deep supervision to strengthen gradient flow. However, it typically introduces additional supervision branches and training overhead, leading to increased computational cost. Importantly, these prior works do not directly address TSP.

## 3. Preliminary

### 3.1. The Leaky Integrate and Fire Model

Spiking neurons are the fundamental computational units of SNNs, enabling spatio-temporal information encoding and propagation. In this work, we consider the widely used Leaky Integrate-and-Fire (LIF) model as the canonical

neuron model for our formulation and experiments, whose membrane potential dynamics are:

$$\tau \frac{dV(t)}{dt} = -(V(t) - V_{rest}) + R \cdot I(t), \quad V(t) < V_{th}, \quad (1)$$

where $\tau$ is the time constant, $V(t)$ is the membrane potential at time $t$ and $V_{rest}$ is the resting potential, which is typically set to 0. $R$ is the resistance, $I(t)$ is the input current at time $t$, and $V_{th}$ is the firing threshold. In current network architecture, the input is given by $I(t) = \sum_i w_i' S_i'(t) + b'$, where $w_i'$ denotes the weight from the i-th neuron in the previous layer to the target neuron, $S_i'(t)$ is the spike train received from the i-th neuron in the previous layer, and $b'$ is a bias term. When the membrane potential exceeds $V_{th}$, the neuron emits a spike and its membrane potential is reset. Specifically, the discrete LIF model with a soft-reset mechanism can be written as:

$$\begin{cases} V[t] = \alpha H[t-1] + \sum_i w_i S_i'[t] + b \\ S[t] = \Theta(V[t] - V_{th}), \\ H[t] = V[t] - S[t], \end{cases} \quad (2)$$

where $\alpha = 1 - \frac{1}{\tau}$ is the decay factor and $H[t]$ is the reset membrane potential at timestep $t$. $w_i$ and $b$ are reparameterized versions of $w_i'$ and $b'$, in which the resistance $R$ and time constant $\tau$ are absorbed. $S[t]$ is the spike emitted at timestep $t$ and $\Theta(\cdot)$ denotes the Heaviside step function.

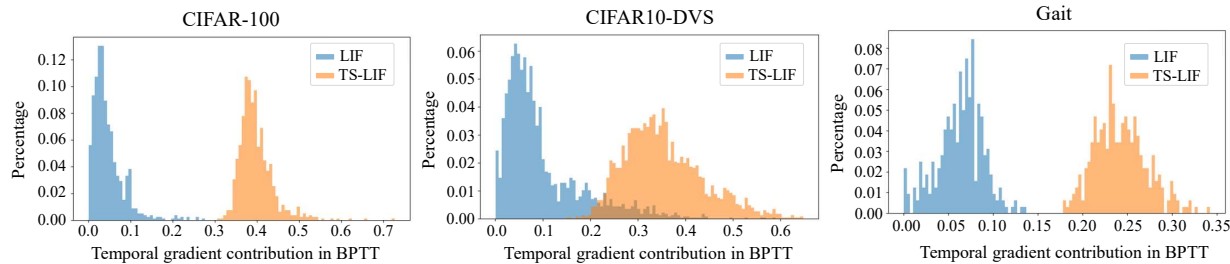

*Figure 3.* Temporal-gradient contribution in BPTT for TS-LIF-based SNN and the vanilla LIF-based SNN. ResNet20 is used for CIFAR-100/CIFAR10-DVS, and 3B-Net is used for Gait. We compute one temporal-gradient contribution per training iteration and report the percentage of samples in each bin. Other relevant details and hyperparameters are provided in Appendix B.

## 3.2. Backpropagation Through Time with Surrogate Gradients

In a multi-layer SNN composed of LIF neurons, the forward propagation can be expressed as:

$$V^l[t] = \alpha H^l[t-1] + W^l S^{l-1}[t], \tag{3a}$$

$$= \alpha(V^l[t-1] - S^l[t-1]) + W^l S^{l-1}[t]. \tag{3b}$$

where $l = 1, 2, \cdots L$ indexes network layers and $t = 1, 2, \cdots, T$ denotes discrete timesteps. For simplicity, we omit bias terms, and $W^l$ denotes the trainable weight matrix of the $l$-th layer. For a classification task, let $o[t]$ denote the network output at timestep $t$. The loss function is defined as

$$\mathcal{L} = \ell(\frac{1}{T} \sum_{t=1}^{T} o[t], y), \tag{4}$$

where $\ell$ is the cross-entropy function and $y$ is the ground-truth class label. According to spatio-temporal backpropagation, the gradient for $W^l$ at timestep $t'$ is:

$$\frac{\partial \mathcal{L}}{\partial W^l} = \sum_{t=1}^{t'} (\frac{\partial \mathcal{L}}{\partial V^l[t]})(S^{l-1}[t])^\top, \tag{5}$$

where

$$\frac{\partial \mathcal{L}}{\partial V^l[t]} = \frac{\partial \mathcal{L}}{\partial S^l[t]} \frac{\partial S^l[t]}{\partial V^l[t]} + \frac{\partial \mathcal{L}}{\partial V^l[t+1]} \frac{\partial V^l[t+1]}{\partial V^l[t]}, \tag{6}$$

and

$$\frac{\partial \mathcal{L}}{\partial S^l[t]} = \frac{\partial \mathcal{L}}{\partial V^{l+1}[t]} \frac{\partial V^{l+1}[t]}{\partial S^l[t]} + \frac{\partial \mathcal{L}}{\partial V^l[t+1]} \frac{\partial V^l[t+1]}{\partial S^l[t]} \tag{7}$$

In Eq. 6 and Eq. 7, $\frac{\partial S^l[t]}{\partial V^l[t]}$ is the **spatial gradient**, which is non-differentiable and thus handled via a surrogate gradient. $\frac{\partial V^l[t+1]}{\partial V^l[t]}$ and $\frac{\partial V^l[t+1]}{\partial S^l[t]}$ are **temporal gradients**.

## 4. Methodology

In this section, we first analyze the long-term temporal gradient issues of the LIF neuron. Then, We then introduce

TS-LIF model, which combines long-term memory reconsolidation and a temporal forgetting mechanism to address these issues. Figure 2 illustrates the overall framework of TS-LIF. Finally, we present a theoretical analysis showing that our method mitigates long-term temporal gradient vanishing and stabilizes temporal gradient flow.

### 4.1. Temporal Gradient Analysis of LIF neuron

Here, we analyze the temporal-term gradients in LIF, as they govern error backpropagation across timesteps under BPTT and are key to explaining TSP.

**Proposition 1.** *For LIF, the $k$-step temporal terms satisfy* $\left\| \frac{\partial V^l[t]}{\partial V^l[t-k]} \right\|_2 \le |\alpha|^k$ *and* $\left\| \frac{\partial V^l[t]}{\partial S^l[t-k]} \right\|_2 \le |\alpha|^k$.

*Proof.* Detailed proofs are provided in Appendix A. $\square$

Since $0 < \alpha < 1$, the temporal terms decay exponentially with $k$, leading to vanishing temporal gradients over long timesteps. Consistent with this analysis, as shown in Figure 3, the temporal-gradient contribution is low for LIF-based SNNs. This makes it difficult for the model to learn how adjusting the current membrane potential $v^l[t]$ and spike $s^l[t]$ can improve future performance at $t+1, \ldots, t+k$, which weakens learning of cross-timestep dependencies. In contrast, our method greatly increases this contribution, enabling more effective learning of temporal dependencies.

### 4.2. Long-term Memory Reconsolidation

To mitigate temporal gradient vanishing and strengthen cross-timestep dependencies, we increase the intrinsic complexity of the spiking neuron such that the membrane potential at the current timestep depends not only on the immediate past state, but also explicitly reuses earlier memory traces. To this end, we introduce a long-term memory reconsolidation, implemented as a temporal residual connection on top of the standard LIF update: at timestep $t$, we additionally inject the membrane potential from timestep $t-2$, with a coefficient $\beta[t]$ controlling the contribution of this

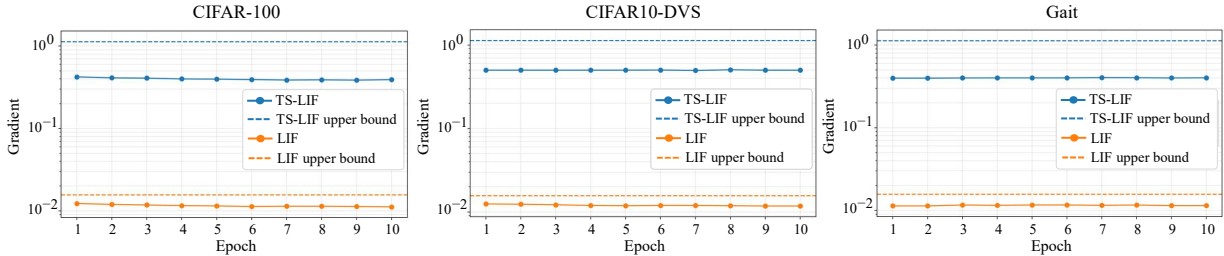

*Figure 4.* Temporal gradient norms for TS-LIF-based SNN and the vanilla LIF-based SNN. Experiments are conducted on CIFAR-100, CIFAR10-DVS, and Gait. We use ResNet20 on CIFAR-100/CIFAR10-DVS and 3B-Net on Gait. We compute the norm of the temporal gradient term across layers and report the average over layers in the figure. Our method mitigates temporal gradient vanishing and does not exhibit gradient explosion in practice. Other relevant implementation details and hyperparameters are reported in the Appendix B.

cross-step signal. Specifically, we rewrite Eq. (3a) as

$$V^l[t] = \begin{cases} \alpha^l H^l[t-1] + W^l S^{l-1}[t], & t \le 2, \\ \alpha^l H^l[t-1] + W^l S^{l-1}[t] \\ \quad + \beta^l[t] V^l[t-2], & t > 2. \end{cases} \quad (8)$$

where $\beta^l[t]$ is a learnable coefficient controlling how much historical membrane trace is reused at timestep $t$ in the $l$-th layer. The long-term memory reconsolidation creates a more direct gradient route across timesteps. Under BPTT, gradients are no longer forced to pass only through the repeatedly attenuated $t-1$ chain scaled by $\alpha$, but can additionally propagate via the $t-2$ shortcut, which improves long-term temporal gradient propagation.

We further make $\alpha$ learnable to allow the network to adapt its temporal decay to the task. Specifically, we replace the shared $\alpha$ in Eq. (8) with a learnable, layer-wise coefficient $\alpha^l$. We adopt a layer-wise $\alpha^l$ rather than a timestep-wise $\alpha^l[t]$ to keep the decay dynamics consistent across time. Timestep-specific modulation is handled by $\beta^l[t]$. Since $\alpha^l$ controls only the short $t-1$ path, overly large values preserve excessive short-range information, leading to redundant accumulation, whereas overly small values can weaken local temporal continuity. Therefore, $\alpha^l$ and $\beta^l[t]$ are learned jointly to balance short decay and long-term injection in a data-driven manner.

### 4.3. Temporal Forgetting Mechanism

After introducing long-term memory reconsolidation, the neuronal state update involves two temporal pathways rather than a single one. As a result, temporal accumulation can become overly aggressive over long timesteps and may amplify temporal gradients, potentially destabilizing optimization during training. To mitigate this issue, we further introduce a temporal forgetting mechanism that periodically truncates the accumulation pathway, regulating long-term temporal build-up and improving optimization stability. Theoretical analysis is provided in Section 4.4.

Specifically, the path $V[t-1] \to V[t]$ is activated at every

timestep, forming a temporally dense carry-over chain. Its influence is repeatedly propagated and compounded along the temporal axis, making it more prone to induce step-wise accumulation and amplification. In contrast, the long-term memory reconsolidation $V[t-2] \to V[t]$ provides a sparse cross-step path and does not create an equally dense per-step propagation chain. Consequently, once memory reconsolidation introduces an additional cross-step path, excessive accumulation is typically carried by the temporally dense path. Yet we do not directly remove it, as it is essential for conveying local temporal continuity between adjacent timesteps. Instead, we introduce a periodic modulation variable $r[t]$ to realize forgetting. When the timestep reaches the forget period $q$, we disable the accumulated contribution from the previous timestep. Eq. (8) can be rewritten as

$$V^l[t] = \begin{cases} r[t]\alpha^l H^l[t-1] + W^l S^{l-1}[t], & t \le 2, \\ r[t]\alpha^l H^l[t-1] + W^l S^{l-1}[t] \\ \quad + \beta V^l[t-2], & t > 2. \end{cases} \quad (9)$$

where

$$r[t] = \begin{cases} 0, & t \equiv 0 \pmod{q}, \\ 1, & \text{otherwise.} \end{cases} \quad (10)$$

Overall, long-term memory reconsolidation provides the cross-step injection needed for long-term interactions, while the temporal forgetting mechanism ensures that such interactions remain well-behaved during long-term training. Together, they complement each other.

In practice, accumulating membrane potentials over timestep and periodically truncating temporal pathways can cause scale drift or numerical instability. Therefore, we apply a normalization operator after the update when $t > 2$ to stabilize the membrane potential distribution. Accordingly, Eq. (9) can be rewritten as

$$V^l[t] = \begin{cases} r[t]\alpha^l H^l[t-1] + W^l S^{l-1}[t], & t \le 2, \\ \mathrm{N}(r[t]\alpha^l H^l[t-1] + W^l S^{l-1}[t] \\ \quad + \beta^l[t] V^l[t-2]), & t > 2. \end{cases} \quad (11)$$

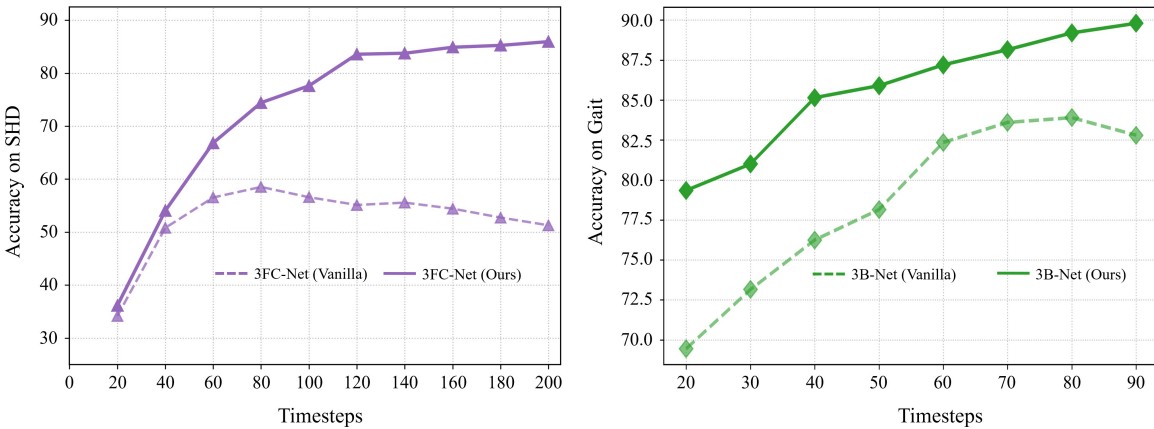

*Figure 5.* Performance trends with scaling timesteps on the SHD and Gait datasets under temporal sampling interval $dt = 5$ and $dt = 10$.

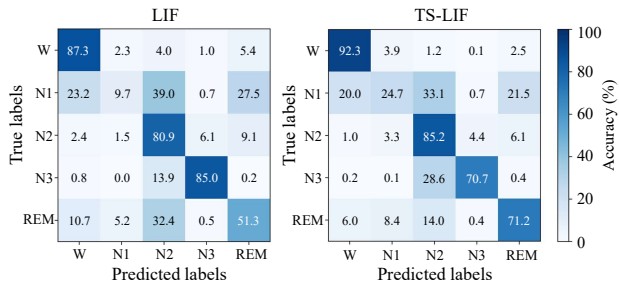

*Figure 6.* Confusion matrices of the LIF neuron and TS-LIF neuron on Sleep-EDF. The overall accuracy is 70.9% vs. 77.2%

$N(\cdot)$ denotes a normalization layer. In this paper, we use GroupNorm (Wu & He, 2018).

### 4.4. Temporal Gradient Analysis of TS-LIF neuron

Here we analyze how TS-LIF affects the temporal gradient.

**Proposition 2.** *Long-term memory reconsolidation mitigates vanishing temporal gradients over long timesteps.*

*Proof.* Detailed proofs are provided in Appendix A. □

**Proposition 3.** *The temporal forgetting mechanism stabilizes temporal gradient propagation and improves training stability over long timesteps.*

*Proof.* Detailed proofs are provided in Appendix A. □

Following the theoretical analysis above, we also empirically examine temporal gradient by measuring the norm of the temporal term $\frac{\partial v^l[t]}{\partial v^l[t-k]}$ for TS-LIF-based SNN and vanilla LIF-based SNN. As shown in Figure 4, TS-LIF yields temporal-term norms that are an order of magnitude larger than those of LIF, substantially alleviates temporal

gradient vanishing, while maintaining stable gradient magnitudes without exhibiting exploding-gradient behavior.

## 5. Experiments

In this section, we first validate the effectiveness of our method under larger numbers of timesteps on the SHD (Cramer et al., 2020) and Gait (Wang et al., 2019) datasets. We then evaluate our method on time-series and event-based datasets, including Sleep-EDF (Goldberger et al., 2000), CIFAR10-DVS (Li et al., 2017), and Electricity (Lai et al., 2018). We also report results on standard vision datasets, including CIFAR-100 (Krizhevsky et al., 2009) and ImageNet (Deng et al., 2009) for image classification, and COCO (Lin et al., 2014) for object detection. CIFAR-100 results are in Appendix D.1. The experimental settings are provided in the Appendix B.

### 5.1. Results on Larger Numbers of Timesteps

To assess the scalability of our method to larger numbers of timesteps, we conduct experiments on the SHD dataset with $dt = 5$ and the Gait dataset with $dt = 10$, as shown in Figure 5. On SHD, our method consistently improves as $T$ increases and outperforms the vanilla baseline by 34.67% at $T = 200$. On Gait, our method attains 89.80% accuracy at $T = 90$, compared to 83.80% for the vanilla baseline. Moreover, our method exhibits a clear monotonic improvement as timesteps increase. In contrast, the vanilla baseline starts to degrade at larger $T$. This demonstrates that our method can scale to larger numbers of timesteps.

### 5.2. Results on Time-series and Event-based Datasets

**Sleep-EDF.** Sleep-EDF is an EEG-based sleep-stage classification datasets. We replace the LIF neuron in Spiking EEGNet (Fan et al., 2025) with TS-LIF neuron and obtain a 6.3% improvement in accuracy on Sleep-EDF-18 (70.9%

*Table 1.* Results on the Electricity dataset. We adopt the same network as CPG-PE. Results are averaged over three random seeds.

| Method | Type | Metric | Electricity | | | | Avg. |
|---|---|---|---|---|---|---|---|
| | | | 6 | 24 | 48 | 96 | |
| Spikformer (Zhou et al., 2023) | BPTT | $R^2 \uparrow$ | .956 | .955 | .953 | .943 | .952 |
| | | RSE $\downarrow$ | .371 | .375 | .386 | .450 | .396 |
| CPG-PE (Lv et al., 2024) | BPTT | $R^2 \uparrow$ | .971 | .971 | .968 | .962 | .968 |
| | | RSE $\downarrow$ | .304 | .308 | .311 | .439 | .341 |
| TS-LIF (Ours) | BPTT | $R^2 \uparrow$ | **.972** | **.974** | **.971** | **.963** | **.970** |
| | | RSE $\downarrow$ | **.298** | **.289** | **.305** | **.343** | **.308** |

*Table 2.* Performance on the CIFAR10-DVS dataset. $T$ denote the number of timesteps.

| Method | Type | Architecture | T | Accuracy (%) |
|---|---|---|---|---|
| Dspike (Li et al., 2021b) | BPTT | ResNet20 | 10 | 75.40 |
| ReverB-SNN (Guo et al., 2025) | BPTT | ResNet20 | 10 | 78.10 |
| FSTA-SNN (Yu et al., 2025a) | BPTT | ResNet20 | 10 | 81.50 |
| RMP-Loss (Guo et al., 2023) | BPTT | ResNet19 | 10 | 76.20 |
| DA-LIF (Zhang et al., 2025) | BPTT | ResNet19 | 10 | 78.00 |
| Ternary Spike (Guo et al., 2024a) | BPTT | ResNet19 | 10 | 79.80 |
| CKA-SNN (Zhang et al., 2024) | BPTT | ResNet20 | 10 | 78.50 |
| | | ResNet19 | 10 | 80.00 |
| TS-LIF (Ours) | BPTT | ResNet20 | 4 | **81.60** $\pm$ 0.16 |
| | | | 10 | **83.20** $\pm$ 0.08 |

vs. 77.2%). The confusion matrices are shown in Figure 6. TS-LIF outperforms the LIF on most sleep stages, demonstrating a clear advantage on physiological time-series tasks.

**Electricity.** We report Electricity results in Table 1. Across a range of prediction lengths (6/24/48/96), our method consistently outperforms Spikformer (Zhou et al., 2023) and CPG-PE (Lv et al., 2024). Notably, the advantage becomes more pronounced at the longest prediction length (96), highlighting its strength in long-range time-series forecasting.

**CIFAR10-DVS.** Results on CIFAR10-DVS are reported in Table 2. With ResNet20, our method achieves 81.60% accuracy at $T = 4$, outperforming prior approaches that use longer timesteps (T=10). Moreover, our method achieves 83.20% accuracy at $T = 10$, substantially exceeding Dspike (75.40%) (Li et al., 2021b) and ReverB-SNN (78.10%) (Guo et al., 2025). These results validate the effectiveness and consistent advantage of our method on neuromorphic datasets.

### 5.3. Additional Results on Standard Vision Datasets

**ImageNet.** Table 3 demonstrates the competitiveness of our method on static image classification. For example, with a ResNet34 architecture, our method achieves 69.46% and 70.11% accuracy at $T = 4$ and $T = 6$, clearly outperforming OTTT (Xiao et al., 2022) and SLTT (Meng et al., 2023) by 4.96% and 3.92%, both of which aim to reduce temporal

gradient propagation during training. This suggests that strengthening the temporal gradient is effective.

**COCO.** Results on COCO are shown in Table 4. Our method achieves the best performance and consistently benefits from increasing timesteps across different model sizes. In contrast, SpikeYOLO (Luo et al., 2024) does not reliably gain from longer timesteps and can even exhibit TSP. For example, with 13.2M parameters, increasing $T$ from 4 to 8 degrades SpikeYOLO (mAP@50: 51.3% to 50.1%, mAP@50:95: 35.3% to 34.7%), while improving ours (mAP@50: 51.8% to 53.6%, mAP@50:95: 35.8% to 37.4%), validating our effectiveness for object detection.

### 5.4. Computational Overhead Analysis

We analyze the computational overhead introduced by TS-LIF compared with vanilla LIF. Specifically, we report the parameter count, training time per epoch, inference time, memory usage, and energy consumption in Table 5. The detailed methodology for energy calculation is provided in Appendix C.

On CIFAR-100, TS-LIF introduces only a negligible parameter increase from 11.29M to 11.30M, with slight increases in training time, inference time, and memory usage. On SHD, under the much longer setting of $T = 200$, TS-LIF similarly increases the parameter count only from 0.109M

*Table 3.* Performance on the ImageNet dataset.

| Method | Type | Architecture | T | Accuracy (%) |
|---|---|---|---|---|
| RMP-SNN (Han et al., 2020) | ANN-SNN | ResNet34 | 512 | 60.08 |
| MS-ResNet (Hu et al., 2024b) | BPTT | ResNet18 | 6 | 63.10 |
| GAC-SNN (Qiu et al., 2024) | BPTT | ResNet18 | 6 | 65.14 |
| OTTT (Xiao et al., 2022) | BPTT | ResNet34 | 6 | 65.15 |
| SLTT (Meng et al., 2023) | BPTT | ResNet34 | 6 | 66.19 |
| RecDis-SNN (Guo et al., 2022) | BPTT | ResNet34 | 6 | 67.33 |
| OSR+OTS (Zhu et al., 2024) | BPTT | ResNet34 | 4 | 67.54 |
| SEW ResNet (Fang et al., 2021) | BPTT | ResNet18 | 4 | 63.18 |
|  |  | ResNet34 | 4 | 67.04 |
| TS-LIF (Ours) | BPTT | ResNet18 | 4 | **66.28** $\pm$ 0.13 |
|  |  | ResNet18 | 6 | **66.98** $\pm$ 0.11 |
|  |  | ResNet34 | 4 | **69.46** $\pm$ 0.08 |
|  |  | ResNet34 | 6 | **70.11** $\pm$ 0.14 |

*Table 4.* Performance comparison on the COCO dataset. We adopt the same backbone architecture as SpikeYOLO.

| Method | Type | Param(M) | T | mAP@50(%) | mAP@50:95(%) |
|---|---|---|---|---|---|
| Spiking-Yolo (Kim et al., 2020) | ANN-SNN | 10.2 | 3500 | - | 25.7 |
| Spike Calib (Li et al., 2022) | ANN-SNN | 17.1 | 512 | 45.4 | - |
| EMS-YOLO (Su et al., 2023) | BPTT | 26.9 | 4 | 50.1 | 30.1 |
| Meta-SpikeFormer (Yao et al., 2024a) | BPTT | 16.8 | 4 | 45.0 | - |
| SpikeYOLO (Luo et al., 2024) | BPTT | 13.2 | 4 | 51.3 | 35.3 |
|  |  |  | 6 | 51.0 | 35.2 |
|  |  |  | 8 | 50.1 | 34.7 |
|  |  | 23.1 | 4 | 55.7 | 38.7 |
|  |  |  | 6 | 55.6 | 39.0 |
|  |  |  | 8 | 55.6 | 39.1 |
| TS-LIF (Ours) | BPTT | 13.2 | 4 | **51.8** | **35.8** |
|  |  |  | 6 | **52.8** | **36.7** |
|  |  |  | 8 | **53.6** | **37.4** |
|  |  | 23.1 | 4 | **56.1** | **39.3** |
|  |  |  | 6 | **56.9** | **40.2** |
|  |  |  | 8 | **57.8** | **41.0** |

to 0.110M, while maintaining practical training and inference costs. Notably, the estimated energy consumption remains comparable to vanilla LIF on both datasets. These results show that TS-LIF does not introduce unacceptable computational overhead and can serve as a practical drop-in replacement for vanilla LIF.

### 5.5. Ablation Study

Table 6 presents an ablation study of memory reconsolidation, temporal forgetting, and normalization. Overall, memory reconsolidation accounts for the main gain, while temporal forgetting and normalization become most effective when combined with it or with each other.

Specifically, memory reconsolidation alone improves accu-

racy by 2.32% on CIFAR-100 and 1.90% on CIFAR10-DVS. This indicates memory reconsolidation strengthens temporal dependency modeling, which is key for SNNs. Temporal forgetting alone yields a modest gain (0.84% on CIFAR-100 and 0.70% on CIFAR10-DVS), likely by mildly suppressing noisy temporal accumulation over long timesteps. Without memory reconsolidation, its benefit remains limited. Normalization alone reduces accuracy. We suggest it mainly stabilizes the membrane potential dynamics induced by reconsolidation and forgetting. In isolation, it introduces an unnecessary rescaling of membrane potentials for $t > 2$.

When combined with memory reconsolidation, temporal forgetting brings additional gains of 1.21% on CIFAR-100 and 0.80% on CIFAR10-DVS, suggesting it suppresses excessive temporal accumulation once a long-term pathway

*Table 5.* Computational overhead comparison between vanilla LIF and TS-LIF. Results are reported as vanilla LIF / TS-LIF.

| Dataset | $T$ | Params (M) | Training Time (s) | Inference Time (s) | Memory (MB) | Energy (mJ) |
|---------|-----|------------|-------------------|--------------------|-------------|-------------|
| CIFAR-100 | 4 | 11.29 / 11.30 | 69 / 74 | 3.55 / 3.83 | 2302 / 2784 | 0.21 / 0.19 |
| SHD | 200 | 0.109 / 0.110 | 38 / 45 | 6.23 / 7.11 | 790 / 844 | 0.0836 / 0.0839 |

*Table 6.* Ablation of memory reconsolidation (R), temporal forgetting (F), and normalization (N) at $T = 10$ with ResNet20.

| Dataset | R | F | N | Accuracy (%) |
|---------|---|---|---|--------------|
| | ✗ | ✗ | ✗ | 72.01 |
| | ✓ | ✗ | ✗ | 74.33 |
| | ✗ | ✓ | ✗ | 72.85 |
| CIFAR-100 | ✗ | ✗ | ✓ | 69.95 |
| | ✓ | ✓ | ✗ | 75.54 |
| | ✓ | ✗ | ✓ | 74.68 |
| | ✗ | ✓ | ✓ | 73.30 |
| | ✓ | ✓ | ✓ | **76.28** |
| | ✗ | ✗ | ✗ | 79.60 |
| | ✓ | ✗ | ✗ | 81.50 |
| | ✗ | ✓ | ✗ | 80.30 |
| CIFAR10-DVS | ✗ | ✗ | ✓ | 78.70 |
| | ✓ | ✓ | ✗ | 82.30 |
| | ✓ | ✗ | ✓ | 81.80 |
| | ✗ | ✓ | ✓ | 81.20 |
| | ✓ | ✓ | ✓ | **83.20** |

is available. Normalization provides gains when combined with other components. For instance, with temporal forgetting, it improves F-only by 0.45% on CIFAR-100 and 0.90% on CIFAR10-DVS, likely by stabilizing the abrupt membrane potential shifts introduced by forgetting. The full model achieves the best performance, reaching 76.28% on CIFAR-100 and 83.20% on CIFAR10-DVS, supporting the complementarity among the three components.

In addition, further ablations on the initialization of $\beta[t]$, the forgetting period $q$, the memory reconsolidation term, the dense memory reconsolidation, the forgetting term, and the forgetting strategy are provided in Appendix D.2. We also analyze the role of temporal gradients for timestep scaling in Appendix D.3.

## 6. Conclusion

We present the first analysis of the counter-intuitive timestep-scaling breakdown in SNNs. We suggest that long-term temporal gradient vanishing is the primary driver. To address this issue, we propose timestep-scalable LIF (TS-LIF), which incorporates long-term memory reconsolidation and a temporal forgetting mechanism. Specifically, TS-LIF reuse earlier membrane potential and periodically truncate accu-

mulation along the decay pathway, improving temporal gradient propagation and stabilizing long-term training, thereby enabling consistent performance gains as timesteps increase. These improvements are supported by theoretical analysis and experimental validation. Moreover, comprehensive results demonstrate the effectiveness of our method across diverse network architectures and tasks. We believe this work opens an alternative path for scaling SNN performance.

## Acknowledgements

The work is supported by Strategic Priority Research Program of the Chinese Academy of Sciences (Grant XDB0930000), Guangdong Provincial Key Laboratory of Multimodality Non-Invasive Brain-Computer Interfaces (Grant No.2024B1212010010), the Natural Science Foundation of China (U2441241, 62536002, U24A20331, 62192782, 62532015), and Beijing Natural Science Foundation (L251005, L252032, L223003).

## Impact Statement

This paper presents work whose goal is to advance the field of Machine Learning. There are many potential societal consequences of our work, none which we feel must be specifically highlighted here.

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

## A. Temporal Gradient Analysis

**Lemma 1** (Spectral norm of diagonal matrices). *For a diagonal matrix $D = \mathrm{diag}(d_1, \ldots, d_n) \in \mathbb{R}^{n \times n}$, we have*

$$\|D\|_2 = \max_{1 \leq i \leq n} |d_i|. \tag{12}$$

*Proof.* The spectral norm $\|\cdot\|_2$ is the induced matrix norm of the Euclidean vector norm, so the claim follows from Theorem 5.6.36 in (Horn & Johnson, 2012). $\square$

**Lemma 2** (Spectral norm of block-diagonal matrices). *For a block-diagonal matrix*

$$B = \mathrm{blkdiag}(B_1, \ldots, B_m), \qquad B_j \in \mathbb{R}^{n_j \times n_j}, \tag{13}$$

*its spectral norm satisfies*

$$\|B\|_2 = \max_{1 \leq j \leq m} \|B_j\|_2. \tag{14}$$

*Proof.* Applying Eq. (1.1) in (Taghavi et al., 2015) repeatedly yields the result for $B = \mathrm{blkdiag}(B_1, \ldots, B_m)$. $\square$

**Lemma 3** (Sub-multiplicativity of spectral norm). *For any conformable sequence of matrices $\{A_j\}_{j=1}^k$,*

$$\left\| \prod_{j=1}^k A_j \right\|_2 \leq \prod_{j=1}^k \|A_j\|_2. \tag{15}$$

*Proof.* The two-matrix inequality is the submultiplicativity property in Theorem 5.6.2(4) in (Horn & Johnson, 2012). Repeatedly applying it yields $\left\| \prod_{j=1}^k A_j \right\|_2 \leq \prod_{j=1}^k \|A_j\|_2$. $\square$

**Proposition 1.** *For the LIF neuron, the $k$-step temporal terms satisfy $\left\| \frac{\partial V^l[t]}{\partial V^l[t-k]} \right\|_2 \leq |\alpha|^k$ and $\left\| \frac{\partial V^l[t]}{\partial S^l[t-k]} \right\|_2 \leq |\alpha|^k$. Thus, LIF dynamics cause temporal gradients to vanish over long timesteps.*

*Proof.* For the temporal gradient term $\frac{\partial V^l[t+1]}{\partial V^l[t]}$, we define the one-step temporal Jacobian

$$J^l[t] = \frac{\partial V^l[t+1]}{\partial V^l[t]} = \alpha \left( I - \frac{\partial S^l[t]}{\partial V^l[t]} \right) = \alpha(I - G^l[t]) \tag{16}$$

Since the spike function is non-differentiable, $G^l[t]$ is approximated via a surrogate gradient. Using the widely adopted straight-through estimator (STE) (Rathi et al., 2020), we define

$$G^l[t] = \mathrm{diag}(g^l[t]), \tag{17}$$

where $g_i^l[t] = \mathbf{1}(0 \leq V_i^l[t] \leq 1)$ and 0 otherwise. Here, the subscript $i$ indexes the $i$-th neuron ($i = 1, ..., n_l$, where $n_l$ denotes the number of neurons in the $l$-th layer).

Since $G^l[t]$ is diagonal, $J^l[t]$ is also diagonal. Therefore, by Lemma 1, we have

$$\|J^l[t]\|_2 = \max_i |\alpha(1 - g_i^l[t])| \le |\alpha|. \tag{18}$$

Following Eq. (6), the temporal gradient term $\frac{\partial V^l[t+1]}{\partial V^l[t]}$ can be unrolled over $k$ steps. Therefore, we consider the $k$-step derivative $\frac{\partial V^l[t]}{\partial V^l[t-k]}$:

$$\frac{\partial V^l[t]}{\partial V^l[t-k]} = \prod_{j=1}^{k} J^l[t-j]. \tag{19}$$

According to Lemma 3, we have

$$\left\| \frac{\partial V^l[t]}{\partial V^l[t-k]} \right\|_2 \le \prod_{j=1}^{k} \|J^l[t-j]\|_2 \le |\alpha|^k. \tag{20}$$

Since the decay factor satisfies $\alpha < 1$, the upper bound of $\left\| \frac{\partial V^l[t]}{\partial V^l[t-k]} \right\|_2$ decays exponentially with $k$, leading to vanishing gradients along the temporal dimension.

Similarly, the other temporal gradient term $\frac{\partial V^l[t+1]}{\partial S^l[t]}$ can also be unrolled over $k$ steps. Therefore, for $\frac{\partial V^l[t]}{\partial S^l[t-k]}$, we have

$$\frac{\partial V^l[t]}{\partial S^l[t-k]} = \frac{\partial V^l[t]}{\partial V^l[t-k+1]} \frac{\partial V^l[t-k+1]}{\partial S^l[t-k]}, \tag{21}$$

where $\frac{\partial V^l[t-k+1]}{\partial S^l[t-k]} = -\alpha I$. Thus, according to Lemma 3,

$$\left\| \frac{\partial V^l[t]}{\partial V^l[t-k+1]} \frac{\partial V^l[t-k+1]}{\partial S^l[t-k]} \right\|_2 \le \left\| \frac{\partial V^l[t]}{\partial V^l[t-k+1]} \right\|_2$$
$$\cdot \left\| \frac{\partial V^l[t-k+1]}{\partial S^l[t-k]} \right\|_2 \le |\alpha|^{k-1} \cdot |\alpha| = |\alpha|^k. \tag{22}$$

This also leads to vanishing gradients along the temporal dimension.

$\square$

**Proposition 2.** *Long-term memory reconsolidation mitigates vanishing temporal gradients over long timesteps.*

*Proof.* Under the long-term memory reconsolidation, the membrane potential follows:

$$V^l[t] = \alpha^l(V^l[t-1] - S^l[t-1]) + W^l S^{l-1}[t] + \beta^l[t] V^l[t-2], \tag{23}$$

This additional temporal dependency turns the original single-path update into a two-path temporal update. To analyze multi-step temporal gradients, we consider the corresponding Jacobian dynamics and introduce the augmented state

$$\mathbf{Z}^l[t] = \begin{bmatrix} V^l[t] \\ V^l[t-1] \end{bmatrix}. \tag{24}$$

Then the state evolution can be written as

$$\mathbf{Z}^l[t+1] = K^l[t]\mathbf{Z}^l[t] + \begin{bmatrix} W^l S^{l-1}[t+1] \\ 0 \end{bmatrix}. \tag{25}$$

where $K^l[t] = \begin{bmatrix} J^l[t] & P^l[t] \\ I & 0 \end{bmatrix} \in \mathbb{R}^{2n_l \times 2n_l}$ denotes the state transition matrix and $P^l[t] = \frac{\partial V^l[t+1]}{\partial V^l[t-1]} = \beta[t+1]I$. $K^l[t]$ can be permuted into a block-diagonal matrix via a permutation matrix. Specifically, let $\{e_i\}_{i=1}^{n_l}$ be the standard orthonormal basis of $\mathbb{R}^{n_l}$. In the augmented space $\mathbb{R}^{2n_l}$, define

$$u_i = \begin{bmatrix} e_i \\ 0 \end{bmatrix}, \quad v_i = \begin{bmatrix} 0 \\ e_i \end{bmatrix}, \quad i = 1, \dots, n_l. \tag{26}$$

For each $i$, we have:

$$K^l[t] u_i = \begin{bmatrix} J^l[t]e_i \\ e_i \end{bmatrix} = \begin{bmatrix} a_i^l[t] e_i \\ e_i \end{bmatrix} = a_i^l[t] u_i + v_i, \tag{27}$$

$$K^l[t] v_i = \begin{bmatrix} P^l[t]e_i \\ 0 \end{bmatrix} = \begin{bmatrix} \beta^l[t+1] e_i \\ 0 \end{bmatrix} = \beta^l[t+1] u_i, \tag{28}$$

where $a_i^l[t] = \alpha^l(1 - g_i^l[t])$. The current ordered basis of $\mathbb{R}^{2n_l}$ is $[u_1, \dots, u_{n_l}, v_1, \dots, v_{n_l}]$. Let $\Pi \in \mathbb{R}^{2n_l \times 2n_l}$ be the permutation matrix, which is also orthogonal, that reorders this basis into $[u_1, v_1, u_2, v_2, \dots, u_{n_l}, v_{n_l}]$. Then we obtain the orthogonal similarity transformation

$$\Pi^\top K^l[t]\Pi = \text{blkdiag}(M_1^l[t], M_2^l[t], \dots, M_{n_l}^l[t]), \tag{29}$$

which puts $K^l[t]$ into a block-diagonal form with $n_l$ blocks of size $2 \times 2$. Each $2 \times 2$ block is given by

$$M_i^l[t] = \begin{bmatrix} a_i^l[t] & \beta^l[t+1] \\ 1 & 0 \end{bmatrix}, \qquad i = 1, \dots, n_l. \tag{30}$$

Since $\Pi$ is orthogonal, $K^l[t]$ and $\Pi^\top K^l[t]\Pi$ are orthogonally similar and thus share the same singular values ($\|K^l[t]\|_2 = \|\Pi^\top K^l[t]\Pi\|_2$). Moreover, for a block-diagonal matrix, by Lemma 2, we have

$$\|K^l[t]\|_2 = \max_{i=1,\dots,n_l} \|M_i^l[t]\|_2. \tag{31}$$

The spectral norm of $M_i^l[t]$ is the largest singular value:

$$\|M_i^l[t]\|_2 = \sigma_{\max}(M_i^l[t]) = \sqrt{\lambda_{\max}((M_i^l[t])^\top M_i^l[t])}, \tag{32}$$

where

$$(M_i^l[t])^\top M_i^l[t] = \begin{bmatrix} (a_i^l[t])^2 + 1 & a_i^l[t]\beta^l[t+1] \\ a_i^l[t]\beta^l[t+1] & (\beta^l[t+1])^2 \end{bmatrix}, \quad (33)$$

which is symmetric. Hence its largest eigenvalue admits the closed form

$$\lambda_{\max}((M_i^l[t])^\top M_i^l[t]) = \frac{\mathrm{tr}_i^l[t] + \sqrt{(\mathrm{tr}_i^l[t])^2 - 4\det_i^l[t]}}{2}, \quad (34)$$

where

$$\mathrm{tr}_i^l[t] = (a_i^l[t])^2 + 1 + (\beta^l[t+1])^2, \ \det_i^l[t] = (\beta^l[t+1])^2. \quad (35)$$

Since $a_i^l[t] = \alpha^l(1 - g_i^l[t])$ and $|1 - g_i^l[t]| \leq 1$, we have $|a_i^l[t]| \leq |\alpha^l|$, and thus

$$\|M_i^l[t]\|_2^2 \leq \frac{\Delta_i^l[t] + \sqrt{(\Delta_i^l[t])^2 - 4(\beta^l[t+1])^2}}{2}, \quad (36)$$

where $\Delta_i^l[t] = (\alpha^l)^2 + 1 + (\beta^l[t+1])^2$. Combining (31) and (36), we obtain

$$\|K^l[t]\|_2 = \max_i \|M_i^l[t]\|_2 \leq \hat{K}^l[t]. \quad (37)$$

where

$$\hat{K}^l[t] = \sqrt{\frac{\Delta_i^l[t] + \sqrt{(\Delta_i^l[t])^2 - 4(\beta^l[t+1])^2}}{2}}. \quad (38)$$

Moreover, by Lemma 3, the $k$-step Jacobian of the augmented state satisfies

$$\|\frac{\partial \mathbf{Z}^l[t]}{\partial \mathbf{Z}^l[t-k]}\|_2 = \|\prod_{j=1}^{k} K^l[t-j]\|_2 \leq \prod_{j=1}^{k} \hat{K}^l[t]. \quad (39)$$

Let $\overline{Z}_k^l[t] = \frac{\partial \mathbf{Z}^l[t]}{\partial \mathbf{Z}^l[t-k]}$. Partition $\overline{Z}_k^l[t] \in \mathbb{R}^{2n_l \times 2n_l}$ into $n_l \times n_l$ blocks:

$$\overline{Z}_k^l[t] = \begin{bmatrix} D_k^l[t] & E_k^l[t] \\ F_k^l[t] & G_k^l[t] \end{bmatrix}, \quad (40)$$

where $D_k^l[t] = \frac{\partial V_t^l}{\partial V_{t-k}^l}$. Define the selection matrices

$$Q_1 = \begin{bmatrix} I_{n_l} & 0 \end{bmatrix}, \qquad Q_2 = \begin{bmatrix} I_{n_l} \\ 0 \end{bmatrix}. \quad (41)$$

Then $D_k^l[t] = Q_1 \overline{Z}_k^l[t] Q_2$. Since $\|Q_1\|_2 = \|Q_2\|_2 = 1$, we have

$$\left\| \frac{\partial V_t^l}{\partial V_{t-k}^l} \right\|_2 = \|D_k^l[t]\|_2 = \|Q_1 \overline{Z}_k^l[t] Q_2\|_2 \quad (42)$$
$$\leq \|Q_1\|_2 \|\overline{Z}_k^l[t]\|_2 \|Q_2\|_2 = \|\overline{Z}_k^l[t]\|_2.$$

Finally, combining Equation (39), we obtain

$$\left\| \frac{\partial V_t^l}{\partial V_{t-k}^l} \right\|_2 \leq \left\| \overline{Z}_k^l[t] \right\|_2 \leq \prod_{j=1}^{k} \|K^l[t-j]\|_2. \quad (43)$$

Moreover, since $0 < \alpha^l < 1$, it holds that $\hat{K}^l[t] \geq |\alpha^l|$, which implies $\prod_{j=1}^{k} \hat{K}^l[t] \geq |\alpha^l|^k$.

For the other temporal gradient term, we consider the Jacobian $\frac{\partial V_t^l}{\partial S_{t-k}^l}$. By the chain rule, we have

$$\frac{\partial V^l[t]}{\partial S^l[t-k]} = \frac{\partial V^l[t]}{\partial V^l[t-k+1]} \frac{\partial V^l[t-k+1]}{\partial S^l[t-k]}. \quad (44)$$

Taking spectral norms and using Lemma 3 yields

$$\left\| \frac{\partial V^l[t]}{\partial S^l[t-k]} \right\|_2 \leq \left\| \frac{\partial V^l[t]}{\partial V^l[t-k+1]} \right\|_2 \cdot \left\| \frac{\partial V^l[t-k+1]}{\partial S^l[t-k]} \right\|_2. \quad (45)$$

Moreover, the single-step Jacobian satisfies

$$\left\| \frac{\partial V^l[t-k+1]}{\partial S^l[t-k]} \right\|_2 = |\alpha^l|. \quad (46)$$

Combining (43) and (46), we obtain

$$\left\| \frac{\partial V^l[t]}{\partial S^l[t-k]} \right\|_2 \leq \left( \prod_{j=1}^{k-1} \hat{K}^l[t] \right) |\alpha^l|. \quad (47)$$

Similarly, we have $(\prod_{j=1}^{k-1} \hat{K}^l[t])|\alpha^l| \geq |\alpha^l|^k$.

Consequently, the long-term memory reconsolidation alters the contraction behavior of temporal gradients: the multi-step bound is no longer governed solely by $|\alpha^l|^k$, which helps alleviate temporal gradient vanishing and strengthens cross-timestep dependencies. □

**Proposition 3.** *Temporal forgetting mechanism stabilizes temporal gradient propagation and improves training stability over long timesteps.*

*Proof.* After introducing the temporal forgetting mechanism, the state update can be written as

$$\mathbf{Z}^l[t+1] = (K^l[t] \odot R) \mathbf{Z}^l[t] + \begin{bmatrix} W^l S^{l-1}[t+1] \\ 0 \end{bmatrix}, \quad (48)$$

where $\odot$ denotes the Hadamard (element-wise) product and $R = \begin{bmatrix} 0 & I \\ I & I \end{bmatrix}$ is the forgetting matrix.

Overall, the membrane potential dynamics can be summarized as:

$$\mathbf{Z}^l[t+1] = \begin{cases} \mathcal{R}(K^l[t])\,\mathbf{Z}^l[t] \\ \quad + \begin{bmatrix} W^l S^{l-1}[t+1] \\ 0 \end{bmatrix}, & t \equiv 0 \pmod{q}, \\ K^l[t]\,\mathbf{Z}^l[t] \\ \quad + \begin{bmatrix} W^l S^{l-1}[t+1] \\ 0 \end{bmatrix}, & \text{otherwise.} \end{cases}$$
$$(49)$$

where $\mathcal{R}(K^l[t]) = K^l[t] \odot R$. Following the same procedure used for $K^l[t]$, the transition $\mathcal{R}(K^l[t])$ can be permuted into $n_l$ independent $2 \times 2$ blocks

$$M_i^R[t] = \begin{bmatrix} 0 & \beta^l[t+1] \\ 1 & 0 \end{bmatrix}. \tag{50}$$

Its spectral norm equals its largest singular value. Since

$$(M_i^R[t])^\top M_i^R[t] = \begin{bmatrix} 1 & 0 \\ 0 & (\beta^l[t+1])^2 \end{bmatrix}, \tag{51}$$

we have $\|M_i^R[t]\|_2 = \max\{1, |\beta^l[t+1]|\}$, and thus

$$\|\mathcal{R}(K^l[t])\|_2 = \max\{1, |\beta^l[t+1]|\}. \tag{52}$$

Over a temporal window of length $k$, the forget operator is activated periodically every $q$ steps. Let

$$m = |\{\, s \in \{t-k,\dots,t-1\} : s \equiv 0 \pmod{q} \,\}| \tag{53}$$

denote the number of forgetting steps within the interval $\{t-k,\dots,t-1\}$. Clearly, $m \le \lceil k/q \rceil$. For forgetting timesteps, we denote $\hat{K}_{\text{ref}}^l[s] = \|\mathcal{R}(K^l[s])\|_2$.

By Lemma 3, we obtain

$$\|H_k^l[t]\|_2 = \left\|\prod_{j=1}^{k} K^l[t-j]\right\|_2$$
$$\le \left(\prod_{s \in \mathcal{N}} \hat{K}^l[s]\right) \left(\prod_{s \in \mathcal{R}} \hat{K}_{\text{ref}}^l[s]\right), \tag{54}$$

where $\mathcal{R}$ and $\mathcal{N}$ are the index sets of forgetting and non-forgetting timesteps in $\{t-k,\dots,t-1\}$, respectively, with $|\mathcal{R}| = m$ and $|\mathcal{N}| = k - m$.

Using the same block-selection argument as in the memory reconsolidation case, we obtain

$$\left\|\frac{\partial V^l[t]}{\partial V^l[t-k]}\right\|_2 \le \left(\prod_{s \in \mathcal{N}} \hat{K}^l[s]\right) \left(\prod_{s \in \mathcal{R}} \hat{K}_{\text{ref}}^l[s]\right). \tag{55}$$

For the other temporal dependency, by the chain rule we have

$$\frac{\partial V^l[t]}{\partial S^l[t-k]} = \frac{\partial V^l[t]}{\partial V^l[t-k+1]}\, \frac{\partial V^l[t-k+1]}{\partial S^l[t-k]}. \tag{56}$$

Taking spectral norms and using Lemma 3 yields

$$\left\|\frac{\partial V^l[t]}{\partial S^l[t-k]}\right\|_2 \le \left\|\frac{\partial V^l[t]}{\partial V^l[t-k+1]}\right\|_2 \cdot \left\|\frac{\partial V^l[t-k+1]}{\partial S^l[t-k]}\right\|_2. \tag{57}$$

Moreover, we have

$$\left\|\frac{\partial V^l[t-k+1]}{\partial S^l[t-k]}\right\|_2 = |\alpha^l|. \tag{58}$$

Combining (55) and (58), we obtain

$$\left\|\frac{\partial V^l[t]}{\partial S^l[t-k]}\right\|_2 \le \left(\prod_{s \in \mathcal{N}'} \hat{K}^l[s]\right) \left(\prod_{s \in \mathcal{R}'} \hat{K}_{\text{ref}}^l[s]\right) |\alpha^l|, \tag{59}$$

where $\mathcal{R}'$ and $\mathcal{N}'$ are the forgetting and non-forgetting index sets within the interval $\{t-k+1,\dots,t-1\}$.

In particular, whenever $\alpha^l \neq 0$, we have $\hat{K}_{\text{ref}}^l[s] < \hat{K}^l[s]$. Therefore, the temporal forgetting mechanism tightens the worst-case bound of the temporal Jacobian by periodically replacing $\hat{K}^l[s]$ with the smaller $\hat{K}_{\text{ref}}^l[s]$ at forgetting steps. This intermittently suppresses excessive amplification without removing the long-term memory reconsolidation, thereby stabilizing backpropagation through time and reducing the risk of training divergence (exploding gradients). □

**Proposition 4.** *The temporal-gradient lower bound of TS-LIF exceeds the temporal-gradient upper bound of the LIF.*

*Proof.* □

## B. Datasets and Experimental Setup

**SHD.** SHD (Cramer et al., 2020) is a large spike-based audio classification dataset with 20 classes, containing 10,420 audio samples of spoken digits from zero to nine in English and German. On the SHD dataset, We train for 100 epochs with a batch size of 128 using the Adam optimizer, with an initial learning rate of 0.001. We adopt the 3FC-Net architecture following (Yao et al., 2021), set the temporal sampling interval to $dt = 5$, and evaluate the model with up to $T = 200$ timesteps in Figure 5.

**Gait.** Gait (Wang et al., 2019) is an event-based recognition dataset with 20 classes, comprising 4,200 samples collected from 21 volunteers. On the Gait dataset, we train for 100 epochs with a batch size of 32 using the Adam optimizer. The initial learning rate is set to 0.0001 with a weight decay of 0.0005, and a cosine annealing schedule is used to gradually decay the learning rate to 0. we adopt the 3B-Net architecture following (Xue et al., 2026). In Figure 1, we use a temporal sampling interval of $dt = 50$. In Figure 5, we use $dt = 10$ to enable evaluation at higher $T$.

**Sleep-EDF.** Sleep-EDF (Goldberger et al., 2000) is an EEG-based dataset containing 197 polysomnograms, annotated into five sleep stages. It includes 187,596 samples, each with a length of 3,000. On Sleep-EDF, we optimize the model using Adam with a learning rate of 0.001. Training runs for 100 epochs with a batch size of 1024, and the learning rate is annealed to zero using a cosine schedule. All results are reported under 10-fold cross-validation.

**CIFAR10-DVS.** CIFAR10-DVS (Li et al., 2017) is a neuromorphic, event-based variant of CIFAR-10 with 10 classes, comprising 9,000 training images and 1,000 test images. We use SGD with an initial learning rate of 0.1, momentum of 0.9, and a weight decay of 0.0001. We train for 400 epochs with a batch size of 64, and apply a cosine annealing schedule to gradually decay the learning rate to 0.

**Electricity.** The Electricity dataset (Lai et al., 2018) provides hourly power consumption data expressed in kilowatt-hours (kWh). On the Electricity dataset, we use the Adam optimizer with a batch size of 128 and a learning rate of 0.001. We employ early stopping during training, with the patience set to 30.

**CIFAR-100.** CIFAR-100 (Krizhevsky et al., 2009) contains 50,000 training images and 10,000 test images at a size of $32 \times 32$ across 100 classes. The experimental settings are the same as those used for CIFAR10-DVS.

**ImageNet.** ImageNet (Deng et al., 2009) is a large-scale and challenging dataset for image classification task that spans 1,000 classes and contains about 1,300,000 images of size $224 \times 224$, with approximately $1,250,000$ images used for training and $50,000$ used for testing. On ImageNet, we set the batch size to 1024 and train for 320 epochs, while keeping the remaining hyperparameters the same as those used for the CIFAR-100 datasets.

**COCO.** COCO (Lin et al., 2014) is the most widely used dataset for object detection. It covers 80 categories and contains about $118,000$ training images and $5,000$ testing images. On COCO, we use SGD with a batch size of 40 and train for 300 epochs. The initial learning rate is set to 0.01, with momentum 0.937 and weight decay 0.0005. We adopt a linearly scheduled learning-rate decay, so that the learning rate is annealed to $0.004\times$ its initial value at the end of training.

In our method, the decay factor is initialized to 0.25, and the reconsolidation coefficient is initialized to 1. Unless otherwise specified, all ResNet results are reported using the SEW-ResNet architecture (Fang et al., 2021). For Figure 3, we set $T = 4$ to compute the temporal gradient contribution in BPTT. For Figure 4, we set $T = 4$ and measure the temporal gradient as $\frac{\partial v[4]}{\partial v[1]}$.

All experiments are conducted on a system equipped with a 56-core Intel Xeon Gold 6330 2.00GHz CPU and 4 NVIDIA GeForce RTX 4090 GPUs.

## C. Energy Consumption Estimation

We estimate the energy consumption based on 45 nm CMOS technology (Horowitz, 2014), where a multiply-accumulate (MAC) operation consumes approximately $4.6$ pJ, while an accumulate (AC) operation consumes approximately $0.9$ pJ. Since SNNs perform event-driven computation, synaptic operations are activated only when spikes are emitted. Therefore, the energy consumption is estimated by scaling the number of synaptic AC operations with the number of timesteps and the average firing rate.

For a convolutional layer, the estimated energy consumption is given by

$$E_{\text{Conv}} = T \times fr \times F_{\text{out}} \times c_{\text{in}} \times c_{\text{out}} \times k^2 \times E_{\text{AC}}, \quad (60)$$

where $T$ denotes the number of timesteps, $fr$ denotes the average firing rate, $F_{\text{out}}$ denotes the spatial size of the output feature map, $c_{\text{in}}$ and $c_{\text{out}}$ are the numbers of input and output channels, respectively, $k$ is the kernel size, and $E_{\text{AC}} = 0.9$ pJ.

For a fully connected layer, the energy consumption is estimated as

$$E_{\text{Linear}} = T \times fr \times f_{\text{in}} \times f_{\text{out}} \times E_{\text{AC}}, \quad (61)$$

where $f_{\text{in}}$ and $f_{\text{out}}$ denote the input and output feature dimensions, respectively.

## D. Additional Experimental Results

### D.1. Comparison with Prior Methods on CIFAR-100

Results on CIFAR-100 are reported in Table 7. With ResNet20, our method achieves 74.84% and 75.89% accuracy at $T = 4$ and $T = 6$, respectively, outperforming ReverB-SNN (Guo et al., 2025) and FSTA-SNN (Yu et al., 2025a) by 1.56% and 1.40% under the same setting. With ResNet19, our method attains 81.72% and 82.14% accuracy at $T = 4$ and $T = 6$, respectively. It substantially outperforms BKDSNN (T=4) (Xu et al., 2024) and TEBN (T=6) (Duan et al., 2022) by 6.77% and 5.73%, respectively. This further demonstrates the effectiveness of our method.

### D.2. Further Ablation Studies

For the sensitivity analysis of the two key hyperparameters in TS-LIF, i.e., the initialization of $\beta[t]$ and the forgetting period $q$, we conduct experiments on Gait, CIFAR-100, and CIFAR10-DVS under two timestep settings, $T = 6$ and $T = 10$. For the remaining ablation studies, we focus on CIFAR-100 and CIFAR10-DVS with $T = 10$ to further

*Table 7.* Performance on CIFAR100.

| Method | Type | Architecture | T | Accuracy (%) |
|---|---|---|---|---|
| RMP (Han et al., 2020) | ANN-SNN | ResNet20 | 2048 | 67.82 |
| SlipReLU (Jiang et al., 2023) | ANN-SNN | ResNet18 | 128 | 78.55 |
| SLTT (Meng et al., 2023) | BPTT | ResNet18 | 6 | 74.38 |
| ReverB-SNN (Guo et al., 2025) | BPTT | ResNet20 | 4 | 73.28 |
| FSTA-SNN (Yu et al., 2025a) | BPTT | ResNet20 | 4 | 73.44 |
| TS-SNN (Yu et al., 2025b) | BPTT | ResNet20 | 4 | 73.46 |
| RecDis-SNN (Guo et al., 2022) | BPTT | ResNet19 | 4 | 74.10 |
| SLT-TET (Anumasa et al., 2024) | BPTT | ResNet19 | 6 | 74.87 |
| BKDSNN (Xu et al., 2024) | BPTT | ResNet19 | 4 | 74.95 |
| TEBN (Duan et al., 2022) | BPTT | ResNet19 | 6 | 76.41 |
| MI-TRQR (Xue et al., 2026) | BPTT | ResNet19 | 4 | 77.70 |
| LM-H (Hao et al., 2023b) | BPTT | ResNet19 | 4 | 80.31 |
| RMP-Loss (Guo et al., 2023) | BPTT | ResNet20 | 4 | 66.65 |
| | | ResNet19 | 4 | 78.28 |
| TS-LIF (Ours) | BPTT | ResNet20 | 4 | **74.84 ± 0.09** |
| | | | 6 | **75.89 ± 0.06** |
| | | ResNet19 | 4 | **81.72 ± 0.05** |
| | | | 6 | **82.14 ± 0.11** |

*Table 8.* The ablation studies of the initialization of $\beta[t]$. Results are reported as accuracy (%).

| Dataset | T | $\beta[t] = 0.25$ | $\beta[t] = 0.5$ | $\beta[t] = 0.75$ | $\beta[t] = 1.0$ |
|---|---|---|---|---|---|
| Gait | 6 | **74.55** | 74.25 | 74.40 | **74.55** |
| | 10 | **81.55** | 81.00 | 80.05 | 81.20 |
| CIFAR-100 | 6 | 75.68 | 75.40 | 75.84 | **75.89** |
| | 10 | 76.10 | 75.90 | 76.17 | **76.28** |
| CIFAR10-DVS | 6 | 81.50 | 81.20 | 81.60 | **82.30** |
| | 10 | 81.40 | 82.20 | 82.30 | **83.20** |

analyze different design choices. We use 3B-Net for Gait and ResNet20 for CIFAR-100 and CIFAR10-DVS.

**Initialization of $\beta[t]$.** We vary the initialization of $\beta[t]$ in $\{0.25, 0.5, 0.75, 1.0\}$. As shown in Table 8, $\beta[t] = 1.0$ is a robust default choice and is used in our experiments. Moreover, changing the initialization generally does not cause large performance variation.

**Forgetting period $q$.** Table 9 reports the analysis of the forgetting period $q$. Overall, $q = 3$ achieves the best or tied-best performance in all cases. We conjecture that $q$ controls a trade-off between temporal accumulation and stability. Overly frequent forgetting can truncate useful temporal integration, whereas infrequent forgetting allows excessive accumulation that can amplify instability and noise. We set $q = 3$ as the default configuration of TS-LIF unless otherwise specified.

**Memory reconsolidation term.** We investigate the effect of injecting earlier membrane potential into the update of

$v[t]$. As shown in Table 10, using $v[t-2]$ as the memory reconsolidation term consistently yields the best performance among all candidates. On CIFAR100, $v[t-2]$ outperforms $v[t-3], v[t-4], v[t-5]$, and $v[t-6]$ by 0.7%, 0.82%, 0.94%, 1.00%, and 2.98%, respectively. on CIFAR10-DVS, the corresponding gains are 1.60%, 1.00%, 0.70%, 1.40%, and 2.00%. The two-step choice provides more useful temporal cues than more distant alternatives.

**Dense memory reconsolidation.** We investigate whether dense memory reconsolidation brings additional benefits. As shown in Table 11, using only $v[t-2]$ for reconsolidation achieves the best performance on both datasets. In contrast, dense memory reconsolidation does not yield consistent improvements and can slightly reduce accuracy, suggesting that it may introduce redundant temporal information. Therefore, the simple $v[t-2]$ memory reconsolidation is both more concise and empirically optimal.

**Forgetting term.** Table 12 reports an ablation on which term is forgotten. Overall, forgetting $v[t-1]$ performs

*Table 9.* The ablation studies of the forgetting period $q$. Results are reported as accuracy (%).

| Dataset | $T$ | $q = 2$ | $q = 3$ | $q = 4$ | $q = 5$ | $q = 6$ |
|---|---|---|---|---|---|---|
| Gait | 6 | 72.20 | **74.55** | 72.75 | 72.35 | 74.00 |
| | 10 | 80.00 | **81.20** | 80.20 | 80.30 | 80.05 |
| CIFAR-100 | 6 | 73.63 | **75.89** | 75.24 | 74.52 | 74.68 |
| | 10 | 74.66 | **76.28** | 75.85 | 75.40 | 74.99 |
| CIFAR10-DVS | 6 | 81.60 | **82.30** | 82.30 | 81.60 | 82.00 |
| | 10 | 81.60 | **83.20** | 82.30 | 82.30 | 82.20 |

*Table 10.* The ablation studies of memory reconsolidation term.

| Dataset | reconsolidation term | Accuracy |
|---|---|---|
| CIFAR100 | $v[t-2]$ | **76.28%** |
| | $v[t-3]$ | 75.58% |
| | $v[t-4]$ | 75.46% |
| | $v[t-5]$ | 75.34% |
| | $v[t-6]$ | 75.28% |
| CIFAR10-DVS | $v[t-2]$ | **83.20%** |
| | $v[t-3]$ | 81.60% |
| | $v[t-4]$ | 82.20% |
| | $v[t-5]$ | 82.50% |
| | $v[t-6]$ | 81.80% |

*Table 11.* Ablation of the dense memory reconsolidation window size $k'$. The memory reconsolidation term aggregates $\{v[t-2], \ldots, v[t-k']\}$.

| **Dataset** | $k'$ | **Accuracy (%)** |
|---|---|---|
| CIFAR-100 | 2 | **76.28** |
| | 3 | 75.92 |
| | 4 | 75.57 |
| | 5 | 75.79 |
| | 6 | 75.76 |
| CIFAR10-DVS | 2 | **83.20** |
| | 3 | 82.50 |
| | 4 | 82.20 |
| | 5 | 82.70 |
| | 6 | 82.70 |

*Table 12.* The ablation studies of forgetting term. The network is ResNet20 and $T = 10$.

| Dataset | forgetting term | Accuracy (%) |
|---|---|---|
| CIFAR100 | $v[t-1]$ | **76.28** |
| | $v[t-2]$ | 75.59 |
| | $v[t-1]$ and $v[t-2]$ | 75.32 |
| CIFAR10-DVS | $v[t-1]$ | **83.20** |
| | $v[t-2]$ | 81.90 |
| | $v[t-1]$ and $v[t-2]$ | 81.60 |

*Table 13.* Ablation of forgetting strategies in TS-LIF. We use ResNet20 with $T = 10$. Random forgetting stochastically drops the designated term at each timestep with probability $p_1$. Constant forgetting deterministically applies a fixed keep rate $p_2$ at each timestep, mixing the normal state update with a soft reset toward the current input. For $q = 3$, we set $p_1 = p_2 = \frac{1}{3}$.

| **Dataset** | **Forgetting strategy** | **Accuracy (%)** |
|---|---|---|
| CIFAR-100 | Random forgetting | 74.48 |
| | Constant forgetting | 75.12 |
| | Periodic forgetting | **76.28** |
| CIFAR10-DVS | Random forgetting | 82.40 |
| | Constant forgetting | 81.20 |
| | Periodic forgetting | **83.20** |

best on both CIFAR-100 and CIFAR10-DVS. On CIFAR-100, forgetting $v[t-1]$ improves accuracy by 0.69% and 0.96% over forgetting $v[t-2]$ and forgetting both $v[t-1]$ and $v[t-2]$, respectively. On CIFAR10-DVS, the corresponding gains are 1.30% and 1.60%. These results suggest that removing the most recent accumulated state effectively suppresses excessive temporal accumulation, whereas forgetting older states (or multiple states) may discard useful temporal information.

**Forgetting strategy.** Table 13 compares different forgetting strategies for TS-LIF. Across both CIFAR-100 and CIFAR10-DVS, periodic forgetting consistently achieves the best performance, outperforming stochastic random forgetting and deterministic constant forgetting. We also observe a dataset-dependent preference between random and constant forgetting: constant forgetting performs better on CIFAR-100, whereas random forgetting is stronger on CIFAR10-DVS. We hypothesize that the sparse and non-stationary event streams in CIFAR10-DVS benefit from stochastic dropping as a regularizer, improving robustness to bursty temporal noise, while the more stationary dynamics of static images favor a deterministic forgetting rule that provides a stable, controlled decay.

### D.3. Temporal Gradients Are Critical for Timestep Scaling?

To assess whether temporal gradients are essential for timestep scaling, we disable temporal gradient propagation in TS-LIF and evaluate at $T = 4$ and $T = 10$. Table 14 shows that increasing $T$ no longer brings gains and instead leads to a slight drop in accuracy. This diagnostic supports that temporal gradients are an important factor underlying TS-LIF's scalability to larger timesteps.

*Table 14.* Effect of disabling temporal gradients in TS-LIF. The network is ResNet20.

| Dataset | T | Accuracy (%) |
|---|---|---|
| CIFAR-100 | 4 | 73.98 |
| | 10 | 73.87 |
| CIFAR10-DVS | 4 | 81.00 |
| | 10 | 80.40 |

## E. Limitations

TS-LIF may increase training time and memory usage compared with vanilla LIF,. This is mainly because the neuron state needs to maintain and reuse earlier membrane potentials during temporal unfolding. Future work could explore more efficient implementations of TS-LIF, such as parallel temporal computation, memory-efficient state caching, or kernel-level optimization on neuromorphic and GPU hardware.

