# OpenReview forum: "Resolving the Timestep Scaling Paradox in Spiking Neural Networks with a Timestep-Scalable Neuron Model"
_ICML.cc/2026/Conference — ICML 2026 regular_

### Official Review · Reviewer_5uFy · 2026-03-04

**Soundness:** 3
**Presentation:** 3
**Significance:** 4
**Originality:** 3
**Overall Recommendation:** 4
**Confidence:** 4

**Summary:**

This paper proposes TS-LIF, which enhances temporal gradients through $V[t-2]$ memory reconsolidation and periodic forgetting, coupled with normalization for training stability. This mechanism resolves the Timestep Scaling Paradox (TSP), allowing SNN performance to scale continuously as timesteps increase. Effectiveness is well-validated across CIFAR10-DVS, ImageNet, and COCO.

**Compliance With Llm Reviewing Policy:**

Affirmed.

**Final Justification:**

Since the authors addressed my concerns, I am inclined to keep my original positive score.

**Key Questions For Authors:**

1. Could you provide a broader sensitivity analysis for $\beta[t]$ and $q$ across diverse tasks and varying $T$, clarifying the rationale for the default settings?
2. How does TS-LIF impact inference latency, compute, and energy? Can you provide quantitative hardware or simulation metrics?
3. Does TS-LIF still eliminate TSP and maintain monotonic gains when integrated with multi-spike encodings, RNN architectures, or significantly larger models?

**Limitations:**

No. The paper’s "Impact Statement" is a generic disclaimer.

Suggestions:

1. Add a real Limitations section about technical, hardware/efficiency realism, etc.
2. Failure modes & safety-critical deployment.
3. Potential negative societal impact.

**Strengths And Weaknesses:**

**Strengths:**

1. Directly addresses the counter-intuitive performance degradation at larger $T$ (TSP), correctly attributing it to vanishing long-term temporal gradients, and provides a highly effective solution.
2. The combination of cross-step injection ($t-2$), periodic truncation ($q$), and normalization is structurally clear and easily integrated into existing LIF frameworks.
3. The validation spans static images, event streams, and object detection. Models consistently benefit from scaling $T$, and comparisons (e.g., with SpikeYOLO) explicitly demonstrate the successful circumvention of TSP.
4. The ablation studies clearly disentangle the module contributions, showing that reconsolidation (R) drives the primary gain while forgetting (F) and normalization (N) act synergistically.

**Weaknesses:**

1. **Hyperparameter Overhead:** The introduced mechanisms ($\beta[t]$, $q$, and normalization) add hyperparameter tuning costs. A more systematic sensitivity analysis across diverse tasks is needed to prove long-term algorithmic robustness.
2. **Unquantified Inference Costs:** While the modifications primarily alter gradient propagation during training, the paper lacks a direct, quantitative analysis of the resulting inference costs, latency, or energy efficiency on actual neuromorphic hardware.
3. **Unverified Generalizability:** As a fundamental modification to the neuron model, its compatibility and generalizability with other advanced training tricks or alternative spike encodings (e.g., multi-spike, asynchronous coding) remain unverified.

---

> ### Author Rebuttal · Authors · 2026-03-31
>
> We are very grateful for your careful and insightful comments. In the following, we answer your questions one by one.
>
> >*W1&Q1: A broader sensitivity analysis for $\beta[t]$ and q across diverse tasks and varying T is needed to justify the default settings and demonstrate robustness.*
>
> *A1*: We additionally conduct a broader sensitivity analysis of $\beta[t]$ and q on Gait, CIFAR-100, and CIFAR10-DVS under under different timestep settings (T=6, 10). Since $\beta[t]$ is a learnable parameter rather than a fixed manually tuned hyperparameter, we mainly examine the sensitivity to its initialization. Specifically, for $\beta[t]$, we consider the initialization set {0.25, 0.5, 0.75, 1.0}; for q, we consider {2, 3, 4, 5, 6}. The results are summarized below.
>
> |Dataset|T|$\beta[t]$=0.25|$\beta[t]$=0.5|$\beta[t]$=0.75|$\beta[t]$=1.0|
> |---|---|---|---|---|---|
> |**Gait**|6|**74.55**|74.25|74.40|**74.55**|
> ||10|**81.55**|81.00|80.05|81.20|
> |**CIFAR-100**|6|75.68|75.40|75.84|**75.89**|
> ||10|76.10|75.90|76.17|**76.28**|
> |**CIFAR10-DVS**|6|81.50|81.20|81.60|**82.30**|
> ||10|81.40|82.20|82.30|**83.20**|
>
> |Dataset|T|q=2|q=3|q=4|q=5|q=6|
> |---|---|---|---|---|---|---|
> |**Gait**|6|72.20|**74.55**|72.75|72.35|74.00|
> ||10|80.00|**81.20**|80.20|80.30|80.05|
> |**CIFAR-100**|6|73.63|**75.89**|75.24|74.52|74.68|
> ||10|74.66|**76.28**|75.85|75.40|74.99|
> |**CIFAR10-DVS**|6|81.60|**82.30**|**82.30**|81.60|82.00|
> ||10|81.60|**83.20**|82.30|82.30|82.20|
>
> For $\beta[t]$, the results show that $\beta[t]$=1 is a robust default choice, which is also the setting we use, and changing the initialization generally does not cause large performance variation. For q, the results show that q=3 is the best choice. These results support that our default settings are stable across datasets and timestep choices.
>
> We will include this broader sensitivity analysis in the revised manuscript.
>
> ---
>
> >*W2&Q2: The inference latency, compute cost, and energy of TS-LIF are not quantitatively analyzed.*
>
> *A2*: Thank you for this important suggestion. To address this concern, we additionally report the computational overhead of TS-LIF compared with vanilla LIF, including parameter count, training time per epoch, inference time, memory usage, and energy consumption. Energy consumption is estimated based on 45 nm CMOS technology [1] (4.6 pJ/MAC, 0.9 pJ/AC).
>
> In addition to CIFAR-100 (T=4), we also include SHD [2], a spiking speech benchmark, to examine the overhead in a longer-timestep setting (T=200). For SHD, we follow the same architecture as in [3]. The results are shown below.
>
> **All results are reported as vanilla/ours.**
> |Dataset|T|Parameters|Training Time|Inference Time|Memory Usage|Energy|
> |---|---|---|---|---|---|---|
> |CIFAR-100|4|11.29M/11.30M|69s/74s|3.55s/3.83s|2302MB/2784MB|0.21mJ/0.19mJ|
> |SHD|200|0.109M/0.110M|38s/45s|6.23s/7.11s|790MB/844MB|0.0836mJ/0.0839mJ|
>
> The results show that replacing vanilla LIF with TS-LIF introduces negligible parameter overhead and only moderate runtime/memory increase, while keeping energy consumption comparable.
>
> We will include this discussion and the corresponding quantitative overhead analysis in the revised manuscript.
>
> ---
>
> >*W3&Q3: The compatibility and generalizability of TS-LIF beyond the current setting remain unverified, especially with advanced training tricks or alternative spike encodings (e.g., multi-spike).*
>
> *A3*: Thank you for this insightful question. To further examine the generalizability of TS-LIF, we additionally integrate it with Multi-Synaptic Firing (MSF) neuron [4] and evaluate it on Gait under T = 4, 6, 8, 10, 12, 14, 16. The results are summarized below.
>
> |T|MSF|MSF+Ours|
> |---|---|---|
> |4|77.10|77.25|
> |6|79.80|80.35|
> |8|80.80|81.90|
> |10|83.75|84.30|
> |12|83.35|85.35|
> |14|83.30|86.50|
> |16|82.50|88.40|
>
> The results show that, even with multi-spike encoding, TS-LIF still consistently alleviates TSP. This provides evidence that our method is compatible with alternative spike encodings, rather than being limited to a single encoding scheme.
>
> ---
>
> >*Limitation: Suggestions: 1. Add a real Limitations section about technical, hardware/efficiency realism, etc. 2. Failure modes & safety-critical deployment. 3. Potential negative societal impact.*
>
> *A4*: Thank you for these helpful suggestions. We will add a dedicated Limitations section in the revised manuscript. It will discuss technical limitations, including computational overhead, energy consumption, and the lack of validation on real neuromorphic hardware. We will also briefly discuss possible failure modes and the need for careful evaluation before deployment in safety-critical settings.
>
> ---
>
> [1] 1.1 computing's energy problem (and what we can do about it). ISSCC, 2014.
>
> [2] The heidelberg spiking data sets for the systematic evaluation of SNNs. TNNLS, 2020.
>
> [3] Temporal-wise Attention SNNs for Event Streams Classification. ICCV 2021.
>
> [4] A multisynaptic spiking neuron for simultaneously encoding spatiotemporal dynamics. NC, 2025.

---

> > ### Author Rebuttal · Reviewer_5uFy · 2026-04-03
> >
> > Good work. The additional explanations and experiments largely address my concerns, It better to add these into the revised paper.

---

> > > ### Author Response · Authors · 2026-04-04
> > >
> > > Thanks very much for your reply and recognition. We are happy to see that your concerns have been addressed, and we will incorporate these into the revised manuscript.

---

### Official Review · Reviewer_LekX · 2026-03-07

**Soundness:** 4
**Presentation:** 4
**Significance:** 3
**Originality:** 3
**Overall Recommendation:** 5
**Confidence:** 4

**Summary:**

This paper proposes the Timestep-Scalable (TS) neuron model, a leaky integrate-and-fire (LIF) model with newly designed long-term memory reconsolidation and temporal forgetting mechanisms. It addresses the timestep scaling paradox (TSP) in spiking neural networks (SNNs) trained with backpropagation through time and surrogate gradients. Thorough analyses and effective solutions to the TSP are presented. Extensive experiments demonstrate that the proposed method handles the TSP well across different scenarios, enabling SNNs to achieve better performance by scaling time steps.

**Compliance With Llm Reviewing Policy:**

Affirmed.

**Final Justification:**

The authors have added a link to the source code. It improves the reproducibility of the work significantly. So I will keep my promise to raise the scores.

**Key Questions For Authors:**

(1) The experimental results show that the performance of SNNs constructed using TS-LIF improves as the number of time steps increases from 4 to 6. I wonder about the upper limit of the improvement brought by TS-LIF. For example, what would happen if the number of time steps were increased from 4 to 100? Is it possible to conduct experiments on small-scale benchmarks to explore this limits?

(2) The limitations of the proposed model are not discussed in the manuscript. It is suggested that such a discussion be added, particularly regarding the change in time consumption when the vanilla LIF is replaced with TS-LIF in SNNs.

**Limitations:**

No. It is suggested that a paragraph discussing the potential limitations of the proposed TS-LIF be added to the revised manuscript.

**Strengths And Weaknesses:**

# Strengths
* The motivation is meaningful, and the methodology is reasonable. This study has the potential to inspire further research.
* The manuscript is well written and easy to follow. Sufficient figures and visualizations are included.
* Various tasks, including electroencephalogram (EEG) signal analysis, event-based recognition, and time-series forecasting, are introduced to comprehensively validate the authors' conclusions.

# Weaknesses
* The manuscript does not provide a link to the source code, which limits the reproducibility of the work.
* I cannot find discussions of the time or energy efficiency of the SNNs constructed using the proposed TS-LIF. Since TS-LIF is more complex than the vanilla LIF, the proposed model may incur higher computational cost. The authors are encouraged to add a discussion demonstrating that replacing LIF with TS-LIF does not lead to unacceptable time consumption in common scenarios.
* Minor issues in the paper's writing:

The Use of abbreviations and full names: The full name "surrogate gradients" appears again in line 12 and 13 Section 2 (subsection SNNs training with BPTT), even though its abbreviation has already been introduced earlier.

The "W" in "we" in line 2 of Section 4 should not be capitalized because it is not at the beginning of a sentence. The passive voice is recommended for this type of statement.

# Reminder
If the authors are unsure how to add a link to the source code under the double-blind review rule, they may use an anonymous method, such as Anonymous GitHub. **If the authors can include a link to the source code in the revised manuscript, I will immediately improve my recommendation for this work.**

---

> ### Author Rebuttal · Authors · 2026-03-31
>
> We sincerely appreciate the time you have dedicated to reviewing our manuscript and for providing such constructive feedback. We will answer each of your concerns.
>
> >*W1: The manuscript does not provide a link to the source code, which limits the reproducibility of the work.*
>
> *A1*: Thank you for the suggestion. We have prepared an anonymous code repository to support the reproducibility of our work: https://anonymous.4open.science/r/TS-LIF-0912.
>
> Due to time constraints, we have currently organized and uploaded the code for the spiking speech dataset SHD [1], the static image classification dataset CIFAR-100, and the EEG recognition dataset Sleep-EDF. We will continue to organize and release the code for the remaining datasets as soon as possible. Nevertheless, the core implementation of the proposed TS-LIF neuron is already included in the current repository.
>
> ---
>
> >*W2: Computational overhead of TS-LIF.*
>
> *A2*: Thank you for this helpful suggestion. To address this concern, we additionally report the computational overhead of TS-LIF compared with vanilla LIF, including parameter count, training time per epoch, inference time, memory usage, and energy consumption. Energy consumption is estimated based on 45 nm CMOS technology [2], where each MAC is assumed to consume approximately 4.6 pJ and each AC approximately 0.9 pJ.
>
> In addition to CIFAR-100 (T=4), we also include SHD [1], a spiking speech benchmark, to examine the overhead in a longer-timestep setting (T=200). For SHD, we follow the same architecture as in [3]. The results are shown below.
>
> **All results are reported as vanilla/ours.**
> |Dataset|T|Parameters|Training Time|Inference Time|Memory Usage|Energy|
> |---|---|---|---|---|---|---|
> |CIFAR-100|4|11.29M/11.30M|69s/74s|3.55s/3.83s|2302MB/2784MB|0.21mJ/0.19mJ|
> |SHD|200|0.109M/0.110M|38s/45s|6.23s/7.11s|790MB/844MB|0.0836mJ/0.0839mJ|
>
> These results suggest that TS-LIF does not introduce unacceptable computational cost in common scenarios, and remains practical as a drop-in replacement for vanilla LIF. We will include this discussion and the corresponding table in the revised manuscript.
>
> ---
>
> >*W3: Minor issues in the paper's writing.*
>
> *A3*: Thank you for the careful suggestions. We will correct these writing and grammatical issues in the revised manuscript, including introducing abbreviations at their first occurrence, capitalization errors, and wording. We will also carefully proofread the full paper to further improve consistency and ensure accurate presentation.
>
> ---
>
> >*Q1: The experimental results show that the performance of SNNs constructed using TS-LIF improves as the number of time steps increases from 4 to 6. I wonder about the upper limit of the improvement brought by TS-LIF. For example, what would happen if the number of time steps were increased from 4 to 100? Is it possible to conduct experiments on small-scale benchmarks to explore this limits?*
>
> *A4*: Thank you for the question. To further explore the upper limit of the improvement brought by TS-LIF under much longer timesteps, we additionally conduct an extended-timestep analysis on the spiking speech benchmark SHD [1], increasing the number of timesteps from 20 to 200. We follow the same architecture as in [3], and the corresponding results are shown in https://anonymous.4open.science/r/TS_SHD-19F8/SHD.png.
>
> The results show that, on SHD, the performance of SNNs constructed with TS-LIF continues to improve as the number of timesteps increases from 20 to 200, whereas the model with vanilla LIF does not maintain this trend. In particular, at T = 200, our method outperforms vanilla LIF by 34.67%. This result further suggests that TS-LIF remains effective and scalable in much longer-timestep settings. We will include this experiment in the revised manuscript.
>
> ---
>
> >*Q2: The limitations of the proposed model are not discussed in the manuscript. It is suggested that such a discussion be added, particularly regarding the change in time consumption when the vanilla LIF is replaced with TS-LIF in SNNs. & Limitations: It is suggested that a paragraph discussing the potential limitations of the proposed TS-LIF be added to the revised manuscript.*
>
> *A5*: Thank you for the suggestion. We will add a dedicated paragraph in the revised manuscript to discuss the limitations of TS-LIF, including its additional computational overhead, as discussed in A2, and to clarify possible directions for future improvement.
>
> ---
>
> [1] The heidelberg spiking data sets for the systematic evaluation of SNNs. TNNLS, 2020.
>
> [2] 1.1 computing's energy problem (and what we can do about it). ISSCC, 2014.
>
> [3] Temporal-wise Attention SNNs for Event Streams Classification. ICCV 2021.

---

> > ### Author Rebuttal · Reviewer_LekX · 2026-04-06
> >
> > Thanks for the detailed and thoughtful rebuttal. With the response, my concerns have been generally resolved. So, , I will maintain my original score 5.

---

> > > ### Author Response · Authors · 2026-04-06
> > >
> > > Thanks very much for your reply and recognition. We are happy to see that your concerns have been addressed.

---

### Official Review · Reviewer_HjRh · 2026-03-12

**Soundness:** 3
**Presentation:** 3
**Significance:** 3
**Originality:** 3
**Overall Recommendation:** 5
**Confidence:** 4

**Summary:**

This paper studies a timestep scaling paradox in spiking neural networks: although increasing timesteps should provide richer temporal modeling capacity, performance often degrades in practice. The paper attributes this phenomenon to long-term temporal gradient vanishing and weakened cross-timestep dependencies, and propose a long-term memory reconsolidation, together with temporal forgetting and normalization, to improve temporal dependency modeling and stabilize training. The paper includes theoretical analysis and extensive experiments across EEG recognition, DVS, time-series forecasting, image classification, and object detection, showing that the proposed method effectively alleviates the paradox and achieves strong performance.

**Compliance With Llm Reviewing Policy:**

Affirmed.

**Final Justification:**

The authors delivered detailed responses that have completely resolved all the concerns raised in my initial review. On this basis, I have determined my final score.

**Key Questions For Authors:**

1.If a different timestep setting is used (e.g., T=6), does the best forgetting period q change?

2.How does BN compare with GN for normalization in this method?

**Limitations:**

Some ablation studies are not discussed in sufficient depth. Please see the weaknesses and questions above.

**Strengths And Weaknesses:**

Strengths:

1.The paper is well motivated. Prior work in the SNN community has mainly focused on network depth, training techniques, or ANN-to-SNN conversion, while the timestep dimension has been relatively underexplored.

2.The proposed method is simple and practical, without relying on overly complicated module design.

3.The theoretical analysis is supported by empirical results, and together they suggest that TS-LIF can effectively mitigate temporal gradient vanishing.

4.The method is evaluated on a broad range of datasets and tasks, and the experimental results are comprehensive.

Weaknesses：

1.The sensitivity analysis of the forgetting period q could be more complete. Since the current ablation is only reported for T=10, It would be useful to test whether the best q changes under other timestep settings, e.g., T=6.

2.There are a few minor grammar and wording issues, such as in line 212 (“Then, We then introduce TS-LIF model”) and in line 933 (“on CIFAR10-DVS ...”).

---

> ### Author Rebuttal · Authors · 2026-03-31
>
> We sincerely thank you for your careful and insightful comments. Below, we answer your questions one by one.
>
> >*W1: The sensitivity analysis of the forgetting period q could be more complete. Since the current ablation is only reported for T=10, It would be useful to test whether the best q changes under other timestep settings, e.g., T=6.*
>
> *A1*: Thank you for this helpful suggestion. Following your suggestion, we conducted an additional ablation study on the forgetting period q under T=6, The results are shown below:
>
> |Dataset|Forgetting Period q|Accuracy|
> |---|---|---|
> |**CIFAR-100**|2|73.63|
> ||3|**75.89**|
> ||4|75.24|
> ||5|74.52|
> ||6|74.68|
> ||no|74.24|
> |**CIFAR10-DVS**|2|81.60|
> ||3|**82.30**|
> ||4|**82.30**|
> ||5|81.60|
> ||6|82.00|
> ||no|81.90|
>
> The results show that when T=6, setting q=3 remains the best choice overall. Therefore, we recommend q=3 as the default setting. We will include these additional results in the revised manuscript and clarify the default choice of q accordingly.
>
> ---
>
> >*W2: There are a few minor grammar and wording issues, such as in line 212 (“Then, We then introduce TS-LIF model”) and in line 933 (“on CIFAR10-DVS ...”).*
>
> *A2*: Thank you for pointing out these grammar and wording issues. We will correct the mentioned errors in the revised version and carefully proofread the entire manuscript to further fix any other grammar or wording issues.
>
> ---
>
> >*Q1: If a different timestep setting is used (e.g., T=6), does the best forgetting period q change?*
>
> *A3*: Thank you for this question. Please refer to our response to W1 for a detailed discussion.
>
> ---
>
> >*Q2: How does BN compare with GN for normalization in this method?*
>
> *A4*: Thank you for this question. We replaced GN with BN in our method and conducted additional experiments under different timestep settings. The results are shown below:
>
> |Dataset|T|Accuracy|
> |---|---|---|
> |**CIFAR-100**|4|74.36|
> ||6|75.05|
> ||8|76.03|
> ||10|75.76|
> ||12|72.09|
> ||14|74.32|
> ||16|73.42|
> |**CIFAR10-DVS**|4|81.60|
> ||6|81.40|
> ||8|81.60|
> ||10|78.30|
> ||12|81.00|
> ||14|80.70|
> ||16|80.20|
>
> The results show that when BN is used in place of GN, the performance is comparable under smaller timestep settings. However, under larger timestep settings, the performance of BN degrades noticeably, and the timestep scaling paradox becomes evident again.
>
> A possible reason is that BN relies on batch-level statistics, while GN normalizes features within each sample. In TS-LIF, as the number of timesteps increases, the temporal dynamics can make feature distributions more heterogeneous across samples. In this case, batch-level statistics may become less representative for individual samples, whereas GN does not rely on cross-sample statistics in the normalization step and therefore behaves more stably. We will include these results in the revised manuscript and clarify the reason for using GN.

---

> > ### Author Rebuttal · Reviewer_HjRh · 2026-04-03
> >
> > Thank you for the detailed and thoughtful rebuttal. Therefore, I will maintain my original score 5.

---

> > > ### Author Response · Authors · 2026-04-03
> > >
> > > Thanks very much for your reply and recognition. We are happy to see that your concerns have been addressed.

---

### Official Review · Reviewer_sSSX · 2026-03-12

**Soundness:** 2
**Presentation:** 3
**Significance:** 2
**Originality:** 3
**Overall Recommendation:** 4
**Confidence:** 5

**Summary:**

This paper identifies the Timestep Scaling Paradox (TSP) in SNNs, where increasing the number of timesteps leads to marginal improvement or even performance degradation. The authors attribute TSP primarily to long-term temporal gradient vanishing. To address this, the paper proposes the Timestep-Scalable LIF (TS-LIF) neuron model with two mechanisms: (1) long-term memory reconsolidation, which injects the membrane potential from timestep t−2 into the current update to create an additional gradient propagation pathway, and (2) a temporal forgetting mechanism. In addition, GroupNorm is applied to stabilize membrane potential distributions. Theoretical analysis and extensive experiments on EEG, event-based, time-series, image classification, and object detection datasets demonstrate consistent performance gains when scaling timesteps.

**Compliance With Llm Reviewing Policy:**

Affirmed.

**Final Justification:**

Thank you for the response. My remaining concerns have been largely addressed. I hope the rebuttal responses are thoroughly reflected in the revised manuscript. I will raise my score to Weak Accept (4).

**Key Questions For Authors:**

Q1. Evaluation on timestep-demanding applications: Given that resolving TSP is the central claim, can the authors provide large-scale validation on applications that inherently require many timesteps (e.g., long-range time-series forecasting, continuous event-stream processing, speech recognition)?

Q2. Experimental evidence for the causal mechanism: Can the authors present experiments measuring vanilla LIF's temporal gradient norms as T increases (e.g., T=4, 6, 8, 10, 16) and correlating these with performance degradation? The current gradient analyses (Figures 3 and 4) are all at fixed T=4 and thus insufficient to establish that gradient vanishing causes TSP.

Q3. Relationship to LSTM/GRU: Can the authors explicitly discuss how TS-LIF's memory reconsolidation and temporal forgetting differ conceptually from LSTM's cell state and forget gate?

Q4. Computational overhead: What is the wall-clock training/inference time and memory overhead of TS-LIF compared to vanilla LIF.

Q5. Why does dense reconsolidation fail? Table 8 shows that aggregating multiple past states degrades performance. Can the authors explain why this contradicts the premise that richer temporal dependencies improve performance?

**Limitations:**

The authors have not adequately discussed the limitations of their work. The following should be explicitly addressed: (a) the primary evaluation is concentrated on static image classification where scaling timesteps is inherently counterproductive, weakening the practical significance of this work; (b) the causal link between temporal gradient vanishing and TSP is not experimentally established; (c) reconsolidation only slows gradient decay rather than eliminating it; and (d) the computational overhead of additional components is not quantified.

**Strengths And Weaknesses:**

Strengths

S1. Well-motivated problem identification with rigorous theoretical grounding. The TSP phenomenon is clearly demonstrated through Figure 1 across multiple architectures and datasets. Proposition 1 rigorously shows that temporal gradient terms in standard LIF decay as |α|^k, providing a clean mathematical explanation for why scaling timesteps fails. The proofs rely on reasonable assumptions (pointwise spike generation, STE surrogate gradient, soft reset).

S2. Complementary design of reconsolidation and forgetting. Memory reconsolidation creates a bypass gradient pathway to mitigate vanishing, while temporal forgetting periodically truncates the dense pathway to prevent excessive accumulation. The ablation study (Table 5) confirms the complementarity of all components.


Weaknesses

W1. Contradiction between the core claim and evaluation. The paper's central contribution is resolving TSP so that SNNs can benefit from long timesteps. However, a substantial portion of the experimental evaluation is devoted to static image classification (CIFAR-100, ImageNet)—a task that inherently does not require temporal dynamics and long timesteps. In the SNN research, image classification has been trending toward reducing timesteps for energy efficiency. To convincingly demonstrate the value of resolving TSP, large-scale validation on applications that inherently demand much longer timesteps—such as long-range time-series forecasting, continuous event-stream processing, speech recognition, or robotic control—is necessary. While the paper includes results on temporal datasets (Sleep-EDF, Electricity, Gait), their scale and diversity are limited, and the largest timestep experiment (T=90) is conducted on only a single dataset (Gait).

W2. Insufficient experimental evidence for the causal link between temporal gradient vanishing and TSP. The paper separately demonstrates (a) that temporal gradients vanish theoretically as |α|^k (Proposition 1) and (b) that TSP exists (Figure 1), but does not experimentally establish a direct causal connection between the two. Crucially, the temporal gradient analyses in Figures 3 and 4 are all conducted at a fixed T=4. No experiment systematically measures how temporal gradient norms change as T increases from 4 to 10 to 16 in LIF and correlates this with performance degradation. Establishing causality would require: (i) systematic measurement of temporal gradient magnitudes as a function of T, (ii) correlation analysis between gradient magnitude and performance, and (iii) ruling out alternative causes of TSP such as the inherent absence of useful temporal information in static data, optimization difficulty in higher-dimensional spaces, or overfitting to temporal noise. Table 12 shows that disabling temporal gradients in TS-LIF removes the scaling benefit, but this only demonstrates that temporal gradients matter within TS-LIF—it does not directly prove that gradient vanishing is the primary cause of TSP in vanilla LIF.

W3. Limited novelty in the core mechanism. The fundamental idea of introducing temporal skip connections to alleviate gradient vanishing in recurrent-like architectures is well-established in the RNN literature. LSTM networks were specifically designed for this purpose, with the cell state providing a gradient highway across timesteps and the forget gate preventing excessive accumulation. TS-LIF's memory reconsolidation (injecting V[t−2] with learnable β) is conceptually analogous to the temporal skip that LSTM's cell state provides, and the temporal forgetting mechanism functionally corresponds to LSTM's forget gate. While the SNN-specific formulation and gradient bound analysis have value, the underlying insight is not new. The paper should explicitly discuss its relationship to LSTM/GRU gating mechanisms.

W4. Insufficient justification for the choice of t−2 reconsolidation. The choice to inject V[t−2] is justified only empirically (Table 7) without theoretical motivation. The dense reconsolidation ablation (Table 8) shows that aggregating multiple past states hurts performance, yet no explanation is given—which seems to contradict the premise that richer temporal dependencies improve performance.

W5. Overstated claims on time-series forecasting (Table 1). At prediction length 96, R^2 is 0.963 for TS-LIF vs. 0.962 for CPG-PE—a negligible difference. While RSE shows a clearer advantage (0.343 vs. 0.439), the inconsistency between metrics undermines the claim of a "pronounced advantage."

W6. Missing computational overhead analysis. TS-LIF introduces V[t−2] memory storage, learnable α and β, periodic forgetting logic, and GroupNorm, but provides no quantitative comparison of training time, inference time, or memory against vanilla LIF.

W7. No evaluation on Transformer-based SNN architectures.

---

> ### Author Rebuttal · Authors · 2026-03-31
>
> We are grateful to your insightful feedback. We will answer your questions point by point below.
>
> >*W1&Q1&Limitation (a): Lacks experimental validation on tasks that inherently require many timesteps.*
>
> *A1*: We additionally evaluate on SHD [1] (an event-based speech dataset) with T scaled from 20 to 200, using the same architecture as [2] and replacing only the LIF with TS-LIF. As shown in the figure (https://anonymous.4open.science/r/TS_SHD-19F8/SHD.png), LIF first improves and then degrades as T increases, whereas TS-LIF continues to improve. At T=200, TS-LIF outperforms LIF by 34.67%. Static-image datasets are used to verify generalization. These additional results will be added to the revised version.
>
> ---
>
> >*W2&Q2&Limitaion (b): Experimental evidence for the link between temporal gradient vanishing and TSP*
>
> *A2*: Thank you for this helpful suggestion. We measure the temporal-gradient norm G(T)=||dV[T]/dV[1]|| in LIF at different T. The results on Gait and CIFAR-100 are shown below.
>
> |Dataset|T|G(T)|Acc|
> |---|---|---|---|
> |**Gait**|12|1.29e-7|80.75|
> ||14|8.06e-9|79.80|
> ||16|4.75e-10|79.40|
> |**CIFAR-100**|4|1.16e-02|74.05|
> ||6|7.02e-4|72.68|
> ||8|4.34e-5|71.52|
> ||10|2.70e-6|72.01|
> ||12|1.30e-7|70.08|
> ||14|8.02e-9|69.19|
> ||16|6.34e-10|45.30|
>
> *Note: On temporal tasks such as Gait, performance initially improves with T as more complete temporal patterns are revealed, but then degrades beyond a turning point, where performance decline (i.e., TSP) is closely associated with temporal gradient vanishing. Therefore, we focus on the degradation regime (for Gait, T $\ge$ 12).*
>
> For Gait, the Pearson correlation between performance and log(G(T)) is r = 0.972 (p = 0.151). For CIFAR-100, the Pearson correlation is r = 0.716 (p = 0.070), and the Spearman correlation is $\rho$ = 0.964 (p = 4.54e-04).
>
> These results support that TSP in LIF is closely associated with temporal gradients vanishing as T increases. If degradation were mainly due to general optimization difficulty, modifying cross-timestep propagation would not be expected to affect performance trends. We will add these results to the revised version.
>
> ---
>
> >*W3&Q3: Relationship to LSTM/GRU and the mechanistic novelty of TS-LIF*
>
> *A3*: TS-LIF shares a similar high-level goal with LSTM/GRU in improving cross-timestep propagation, but it is conceptually and mechanistically different. LSTM/GRU use continuous hidden/cell states and explicit gates, with the highway established through gated additive state propagation. In contrast, TS-LIF does not introduce an auxiliary cell state or standard recurrent gates, but directly modifies the membrane update under spike/reset dynamics.
>
> ---
>
> >*W4&Q5: Why V[t-2] reconsolidation, and why does dense reconsolidation fail?*
>
> *A4*: V[t-2] already implicitly carries earlier temporal information through recurrent membrane dynamics, so reconsolidation from V[t-2] is sufficient to provide cross-timestep context. Dense reconsolidation instead aggregates highly correlated past membrane states, mainly adding redundancy that can hurt performance.
>
> ---
>
> >*W5: Overstated claim on time-series forecasting performance*
>
> *A5*: Thanks, we will revise “becomes more pronounced” to “shows consistent improvement”. We also evaluate on Solar using QKFormer (following [3]) at prediction length 96. Replacing the LIF with TS-LIF improves performance from R² = 0.698, RSE = 0.564 to R² = 0.711, RSE = 0.552, supporting consistent gains on forecasting tasks.
>
> ---
>
> >*W6&Q4: Computational overhead of TS-LIF*
>
> *A6*: We additionally report the computational cost of TS-LIF on CIFAR-100 (T=4) and SHD (T=200) compared with vanilla LIF, including parameters (P), training time per epoch (TT), inference time (IT), memory (M), and energy (E). Energy is estimated based on 45 nm CMOS technology (4.6 pJ/MAC, 0.9 pJ/AC). Results are shown below.
>
> **All results are reported as vanilla/ours.**
> |Dataset|T|P|TT|IT|M|E|
> |---|---|---|---|---|---|---|
> |CIFAR-100|4|11.29M/11.30M|69s/74s|3.55s/3.83s|2302MB/2784MB|0.21mJ/0.19mJ|
> |SHD|200|0.109M/0.110M|38s/45s|6.23s/7.11s|790MB/844MB|0.0836mJ/0.0839mJ|
>
> TS-LIF introduces negligible parameter overhead and comparable energy, with only moderate increases in runtime and memory. We will include these results in the revision.
>
> ---
>
> >*W7: No evaluation on Transformer-based SNN*
>
> *A7*: Sorry for the confusion. Electricity experiments already use Spikformer (as does CPG-PE). The additional Solar results are based on QKFormer (see A5). Extending TS-LIF to more Transformer-based SNN architectures remains important future work.
>
> ---
>
> >*Limitation(c):  reconsolidation only slows gradient decay rather than eliminating it*
>
> *A8*: Thanks. We will state this explicitly in the limitations section.
>
> ---
>
> [1] The heidelberg spiking data sets for the systematic evaluation of SNNs. TNNLS, 2020.
>
> [2] Temporal-wise Attention SNNs for Event Streams Classification. ICCV 2021.
>
> [3] Toward Relative Positional Encoding in Spiking Transformers. NeurIPS 2025.

---

> > ### Author Rebuttal · Reviewer_sSSX · 2026-04-03
> >
> > Thanks for the authors' response. With the response, my concerns have been partially resolved. The following concerns remain.
> >
> > Q3: Can the authors provide a theoretical or mathematical argument distinguishing TS-LIF from LSTM's cell state and forget gate?
> > Q5: Can the authors provide theoretical or experimental support forr why t−2 is the optimal reconsolidation step, and why dense reconsolidation fails?
> > W7: Can the authors validate TS-LIF on a broader range of Transformer-based SNN architectures with larger-scale benchmarks?

---

> > > ### Author Response · Authors · 2026-04-06
> > >
> > > Thank you again for your careful and insightful comments. Below, we respond to your questions one by one.
> > >
> > > >*Q3: Can the authors provide a theoretical or mathematical argument distinguishing TS-LIF from LSTM's cell state and forget gate?*
> > >
> > > *A9*: Thank you for this helpful suggestion. TS-LIF is mathematically different from LSTM in state, gating, and temporal gradient propagation.
> > >
> > > LSTM introduces an explicit memory state:
> > >
> > > $
> > > c[t] = f[t] \odot c[t-1] + i[t] \odot \tilde{c}[t],\qquad
> > > h[t] = o[t] \odot \phi(c[t]),
> > > $
> > >
> > > where $c[t]$ is the cell state and $h[t]$ the hidden state. The forget gate $f[t]$, input gate $i[t]$, output gate $o[t]$, and the candidate state $\tilde{c}[t]$ are computed from the current input $x[t]$ and previous hidden state $h[t-1]$. Thus, LSTM relies on a dedicated memory variable $c[t]$, and its temporal retention is controlled by continuous, input-dependent gates.
> > >
> > > By contrast, TS-LIF modifies spiking membrane dynamics (for simplicity, we omit the normalization operator):
> > >
> > > $
> > > V^l[t] = r[t]\alpha^l(V^l[t-1]-S^l[t-1]) + W^l S^{l-1}[t] + \beta^l[t]V^l[t-2],
> > > $
> > >
> > > with
> > >
> > > $
> > > r[t] =
> > > \\begin{cases}
> > > 0, & t \\equiv 0 \\pmod{q} \\\\
> > > 1, & \\text{otherwise}
> > > \\end{cases}
> > > $
> > >
> > > This leads to three key differences.
> > >
> > > 1.**No dedicated memory state.** TS-LIF introduces no auxiliary variable like $c[t]$. $V[t-2]$ is a reused historical membrane potential in the same dynamics, not a separate memory state.
> > >
> > > 2.**No data-dependent forget gate.** $r[t]$ is not equivalent to the LSTM forget gate $f[t]$. $f[t]$ is a continuous, input-dependent gate on $c[t]$, whereas $r[t]$ is a fixed periodic binary truncation on the adjacent carry-over path. It is not data-dependent and does not suppress the reconsolidation term.
> > >
> > > 3.**Different temporal gradient mechanism.** In LSTM, long-range gradients are mainly carried by the cell-state chain:
> > >
> > > $
> > > \frac{\partial c[t]}{\partial c[t-k]} = \prod_{j=t-k+1}^{t} f[j],
> > > $
> > >
> > > so its gradient highway is governed by gated recurrence through the dedicated memory state.
> > >
> > > By contrast, in TS-LIF, the temporal gradient is determined by factor $r[t]$, the adjacent path, and the reconsolidation path:
> > >
> > > $
> > > \frac{\partial V[t]}{\partial V[t-k]}=r[t]\alpha \frac{\partial (V[t-1]-S[t-1])}{\partial V[t-k]}+\beta[t]\frac{\partial V[t-2]}{\partial V[t-k]}.
> > > $
> > >
> > > Thus, TS-LIF does not form an LSTM-style memory highway, but improves cross-timestep propagation through an added shortcut and periodic truncation.
> > >
> > > Therefore, TS-LIF is better viewed as modified spiking membrane dynamics rather than an LSTM-like memory-gating architecture.
> > >
> > > ---
> > >
> > > >*Q5: Can the authors provide theoretical or experimental support forr why t−2 is the optimal reconsolidation step, and why dense reconsolidation fails?*
> > >
> > > *A10*: Thanks for this insight question. To understand why dense reconsolidation fails while t-2 is effective, we measure mutual information (MI) between timestep-wise representations as a proxy for temporal redundancy [1]. We compare t-2 and dense reconsolidation (window size 6), and report average off-diagonal and long-lag MI (lag-k: average MI between (t,t-k)):
> > >
> > > |Dataset|Method|Off-diagonal MI|Lag-3|Lag-4|Lag-5|Lag-6|
> > > |---|---|---|---|---|---|---|
> > > |**CIFAR-100**|t-2|0.175|0.161|0.200|0.143|0.128|
> > > ||dense|0.213|0.255|0.210|0.167|0.137|
> > > |**CIFAR10-DVS**|t-2|0.216|0.214|0.250|0.168|0.164|
> > > ||dense|0.282|0.321|0.289|0.227|0.185|
> > >
> > > Dense reconsolidation yields higher off-diagonal and long-lag MI, indicating stronger cross-timestep redundancy, which degrades performance.
> > >
> > > By contrast, t-2 introduces a non-adjacent shortcut that strengthens cross-timestep dependency while avoiding excessive redundancy. We do not claim t-2 is optimal among all lags. Our results support it as a simple, effective choice balancing temporal dependency and redundancy.
> > >
> > > ---
> > >
> > > >*W7: Can the authors validate TS-LIF on a broader range of Transformer-based SNN architectures with larger-scale benchmarks?*
> > >
> > > *A11*: Thank you for the valuable suggestion. Due to the limited rebuttal time, we additionally evaluate TS-LIF on a representative Transformer-based SNN, Spikformer [2], on the large-scale ImageNet benchmark with T=4. Results are below:
> > >
> > > |Model|Acc|
> > > |---|---|
> > > |Spikformer|70.24 |
> > > |Spikformer + Ours|71.19|
> > >
> > > TS-LIF improves accuracy by 0.95%, indicating its effectiveness on Transformer-based SNNs at larger scale. While this is not yet a broad evaluation, it demonstrates that TS-LIF generalizes beyond CNN-based SNNs. We will include more Transformer-based results in the revision.
> > >
> > > ---
> > >
> > > [1] MI-TRQR: Mutual Information-Based Temporal Redundancy Quantification and Reduction for Energy-Efficient SNNs. NeurIPS, 2025.
> > >
> > > [2] Spikformer: When Spiking Neural Network Meets Transformer. ICLR, 2023.

---

### Decision · Program_Chairs · 2026-04-30

**Decision:**

Accept (regular)

**Comment:**

**Summary of reviews.** Four reviewers assess the paper positively (two Accept at 5, two Weak Accept at 4). The paper identifies and names the "Timestep Scaling Paradox" (TSP): increasing the number of simulation timesteps T — expected to improve performance — can paradoxically degrade it due to temporal gradient vanishing. The proposed TS-LIF neuron model introduces memory reconsolidation (cross-timestep information injection at t-2) and temporal forgetting to maintain gradient flow. Results span EEG, event cameras, time-series forecasting, and static image classification.

**Key strengths.**
- Well-motivated and clearly demonstrated paradox — Figure 1 and Proposition 1 (gradient decay as |α|^k) are compelling (consensus across all 4 reviewers).
- Simple and practical neuron model: the TS-LIF drop-in replacement for LIF is easy to implement and broadly applicable (Reviewer HjRh, 5uFy).
- Broad experimental coverage: static images, event streams, EEG, time-series forecasting, and object detection (Reviewer HjRh, 5uFy).
- Strong rebuttal with new experimental evidence: SHD long-timestep scaling (T=20→200, TS-LIF outperforms LIF by 34.67% at T=200), causal gradient analysis with Pearson/Spearman correlations, Spikformer integration on ImageNet (+0.95%), multi-spike compatibility, overhead analysis.

**Key weaknesses and rebuttal assessment.**
- Evaluation on static image classification doesn't truly require long timesteps (Reviewer sSSX, the most critical, confidence 5/5) → authors added SHD (T up to 200) showing clear TSP in temporal data. **Largely resolved (sSSX acknowledged in final response).**
- Novelty concern: reconsolidation/forgetting mechanisms resemble LSTM cell state/forget gate (Reviewer sSSX) → authors provided detailed equation-level comparison identifying three key structural differences (no dedicated memory state, no data-dependent forget gate, different gradient mechanism). **Partially resolved; sSSX remained somewhat skeptical but raised score.**
- Why t-2 reconsolidation specifically? (Reviewer sSSX) → mutual information analysis showing dense reconsolidation increases cross-timestep redundancy. **Addressed.**
- Missing computational overhead analysis (Reviewer sSSX, LekX) → provided parameter count, training/inference time, memory, and energy comparisons. Overhead is moderate. **Resolved.**
- No source code (Reviewer LekX) → anonymous code repository added during rebuttal. LekX explicitly stated they would raise their score upon code release, and confirmed in final justification. **Resolved (score raised).**
- Hyperparameter sensitivity and broader settings (Reviewer 5uFy) → tested beta and q across 3 datasets at T=6 and T=10; defaults robust. **Resolved.**

**Discussion outcome.** Three reviewers (HjRh, LekX, 5uFy) marked concerns as fully resolved. The most critical reviewer (sSSX, confidence 5/5) initially marked concerns as "partially resolved" but after a second round of author responses posted a "Final Response" acknowledging remaining concerns were "largely addressed" and confirmed their Weak Accept (4) score. All reviewers expect rebuttal additions to appear in the camera-ready.